# Turning Challenges into Opportunities: How Distribution Shifts Enhance Identifiability in Causal Representation Learning

## Abstract

Causal representation learning seeks to uncover latent causal variables and their relationships from observed, unstructured data, a task complicated by identifiability challenges. While distribution shifts, viewed as natural interventions on latent causal variables, often present difficulties in traditional machine learning tasks, they also create valuable opportunities for identifiability by introducing variability in latent variables. In this paper, we study a non-parametric condition characterizing the types of distribution shifts that contribute to identifiability within the context of latent additive noise models. We also present partial identifiability results when only a portion of distribution shifts meets the condition. Furthermore, we extend our findings to latent post-nonlinear causal models. Building on our theoretical results, we propose a practical algorithm facilitating the acquisition of reliable latent causal representations. Our algorithm, guided by our underlying theory, has demonstrated outstanding performance across a diverse range of synthetic and real-world datasets. The empirical observations closely align with the theoretical findings, affirming the robustness and effectiveness of our proposed approach.

## 1 Introduction

Causal representation learning holds the promise of identifying pivotal latent causal variables that govern a system's behavior, as well as the intricate causal relationships among them (Schölkopf et al., 2021). By uncovering the underlying causal structure, this field not only enhances the interpretability of models but also improves their ability to generalize to new, unseen data arising from intervention (Peters et al., 2017; Pearl, 2000; Spirtes et al., 2001). Despite these advantages, the foundational theories, particularly concerning identifiability, *e.g.*, the uniqueness of causal representations, present a complex and nuanced challenge.

From a causal representation perspective, distribution shifts can be interpreted as natural interventions acting on latent causal variables rather than on observed variables. This is because causal representation learning typically focuses on causal relationships arising from interactions among latent variables. Such shifts frequently occur across diverse fields, including medical imaging (Chandrasekaran et al., 2021), biogeography (Pinsky et al., 2020), and finance (Gibbs & Candes, 2021). While these distribution shifts often pose challenges in machine learning tasks, e.g., domain generalization and adaptation, they also offer valuable opportunities for identifiability analysis. By comparing different distributions, we may gain insights into which latent variables change and which remain unchanged. This asymmetric information about the variability of latent variables helps in identifying both the latent variables and the associated graph structures. Ultimately, comparative analysis sheds light on the underlying causal mechanisms governing the relationships between latent variables, making the investigation of distribution shifts a promising approach for identifiability.

A critical question arises when leveraging distribution shifts for the identifiability of causal representations: What types of distribution shifts contribute to identifiability? Broadly, two primary categories exist for modeling these shifts: those arising from hard interventions (von Kügelgen et al., 2023; Brehmer et al., 2022; Ahuja et al., 2023; Seigal et al., 2022; Buchholz et al., 2023; Varici et al., 2023) and those from soft interventions (Liu et al., 2022; Zhang et al., 2023; Liu et al., 2024)

[1]. To understand the distinction between these two approaches, it is important to recognize that distribution shifts can be viewed as consequences of interventions *initiated by nature* (Rosenzweig & Wolpin, 2000; Huang et al., 2019; Huang* et al., 2020). In other words, distribution shifts often arise from *self-initiated behaviors* within a causal system. This perspective is particularly relevant in latent spaces, where latent causal variables are unobservable. Hard interventions, however, require that these self-initiated behaviors follow specific patterns, such as assigning fixed values to a latent variable. This constraint can be limiting, as these behaviors are typically arbitrary and uncontrollable. In contrast, soft interventions offer more flexibility by accommodating a wider range of self-initiated behaviors, such as applying functional transformations to a latent variable, making them a more adaptable framework for modeling self-initiated behaviors (Rosenzweig & Wolpin, 2000). However, prior work has been limited to parametric models, focusing primarily on linear or polynomial (Liu et al., 2022; 2024). ~~Given space constraints,~~ See Section 2 for additional discussions.

This work investigates the distribution shifts induced by soft interventions for achieving identifiability in general latent additive noise models. Additive noise models are particularly valuable in modern deep learning due to their simplicity and compatibility with flexible network architectures, especially when compared to polynomial models. For example, the nonlinear component of additive noise models can be effectively implemented using architectures such as Multilayer Perceptrons (MLPs) and transformers (Vaswani, 2017), enabling more robust and adaptable modeling of causal dynamics across diverse scenarios. By introducing a non-parametric condition that characterizes the types of distribution shifts, and building on assumptions from nonlinear ICA (Hyvarinen & Morioka, 2016; Hyvarinen et al., 2019; Khemakhem et al., 2020; Sorrenson et al., 2020), we demonstrate that latent additive noise causal models can be identified up to trivial permutation transformations with scaling. Furthermore, we extend our analysis to practical scenarios where only a subset of the data with distribution shifts meets the specified condition, resulting in partial identifiability. Crucially, this partial identifiability implies that the proposed condition for characterizing distribution shifts is *sufficient and necessary* for identifiability, without requiring additional assumptions, under the framework of nonlinear ICA. We further generalize our identifiability results from latent additive noise causal models to latent post-nonlinear causal models, which are more flexible and encompass additive noise models as a special case. To validate our findings, we have developed a novel method for learning latent additive noise causal models. Empirical experiments on synthetic data, image datasets, and real fMRI data demonstrate the robustness and effectiveness of our proposed approach, aligning closely with the theoretical identifiability results.

## 2 RELATED WORK

Given the challenges associated with identifiability in causal representation learning, numerous existing works tackle this issue by introducing specific assumptions. We categorize these related works into three primary parts based on the nature of these assumptions.

**Special graph structure** Some progress in achieving identifiability centers around the imposition of specific graphical structure constraints (Silva et al., 2006; Shimizu et al., 2009; Anandkumar et al., 2013; Frot et al., 2019; Cai et al., 2019; Xie et al., 2020; 2022a; Lachapelle et al., 2021). Essentially, these graph structure assumptions reduce the space of possible latent causal representations or structures, by imposing specific rules for how variables are connected in the graph. One popular special graph structure assumption is the presence of two pure children nodes for each causal variable (Xie et al., 2020; 2022a; Huang et al., 2022). Very recently, the work in (Adams et al., 2021) provides a viewpoint of sparsity to understand previous various graph structure constraints.However, any complex causal graph structures may appear in real-world scenarios, beyond the pure sparsity assumption. In contrast, our approach adopts a model-based representation for latent variables, allowing arbitrary underlying graph structures.

**Temporal Information** The temporal constraint that the effect cannot precede the cause has been applied in causal representation learning (Yao et al., 2021; Lippe et al., 2022b; Yao et al., 2022; Lippe et al., 2022a). The success of utilizing temporal information to identify causal representations can be attributed to its innate ability to establish causal direction through time delay. By tracking the

---

[1]A hard intervention fixes a variable's value or removes its parent edges, while a soft intervention modifies its distribution, usually retaining parent edges but encompassing hard interventions as a special case. See Appendix K for further details.

sequence of events over time, we gain the capacity to infer latent causal variables. In contrast to these approaches, our focus lies on discovering instantaneous causal relations among latent variables.

**Interventional Data** Exploring distribution shifts for identifying causal representations has been significantly developed recently (Von Kügelgen et al., 2021; Liu et al., 2022; Brehmer et al., 2022; Ahuja et al., 2023; Seigal et al., 2022; Buchholz et al., 2023; Varici et al., 2023; von Kügelgen et al., 2023). The key question is how to model the types of distribution shifts contributing to identifiability. The majority of works focus on using hard interventions to capture the types of distribution shifts, with some specifically considering single-node hard interventions (Ahuja et al., 2023; Seigal et al., 2022; Buchholz et al., 2023; Varici et al., 2023). However, hard interventions may only capture the specific types of distribution shifts. In contrast, soft interventions offer the potential to model a wider array of distribution shifts (Liu et al., 2022; 2024). Unfortunately, the work in Liu et al. (2022) assumes the underlying causal relations among latent causal variables to be linear models, the work in Liu et al. (2024) explores distribution shifts in the context of latent polynomial models, which are susceptible to issues such as numerical instability and exponential growth in terms (Press, 2007; Hastie et al., 2009; Bishop & Nasrabadi, 2006). In this work, we explores distribution shifts in general latent additive noise models, and extend it to more powerful latent post-nonlinear models. This marks a significant advancement over the polynomial models in Liu et al. (2024). It not only avoids issues like numerical instability and the exponential growth associated with polynomial models but also enables the use of non-parametric models, such as MLPs and transformers. This is particularly important, as the success of modern machine learning heavily relies on such complex network architectures. This work also differs from the recent study by Zhang et al. (2023) in several ways. While the latter assumes the mixing function from latent causal variables to observational data is a full row rank polynomial—a constraint that may be limiting in real-world applications—we impose no such restriction. Furthermore, Zhang et al. (2023) requires single-node interventions, where an intervention on each latent node is available. This requirement may be particularly limiting, especially when considering the distribution shifts resulting from self-initiated behaviors within a causal system. In contrast, our approach does not necessitate single-node interventions.

# 3 IDENTIFIABLE LATENT ADDITIVE NOISE MODELS

In this section, we show that by leveraging distribution shifts, latent additive noise models with noise sampled from two-parameter exponential causal representations are identifiable, which also implies that the corresponding latent causal structures can be recovered. We begin by introducing our defined latent additive noise causal models in Section 3.1, aiming to facilitate comprehension of the problem setting and highlight our contributions. Following this, in Section 3.2, we present our identifiability result by establishing a sufficient and necessary condition that characterizes the types of distribution shifts, under common assumptions used in nonlinear ICA. We, additionally, show partial identifiability results, addressing scenarios where only a portion of distribution shifts is available in Section 3.3. This exploration narrows the gap between our findings and practical applications.

## 3.1 LATENT ADDITIVE NOISE MODELS WITH DISTRIBUTION SHIFTS

In our investigation, we explore the following latent causal generative models ~~that elucidate the underlying processes~~. Within these models, the observed data, represented as $\mathbf{x}$, is generated through latent causal variables denoted as $\mathbf{z}$ (where $\mathbf{z} \in \mathbb{R}^\ell$). Furthermore, these latent causal variables $\mathbf{z}$ are generated by combining latent noise variables $\mathbf{n} \in \mathbb{R}^\ell$, known as exogenous variables in causal systems, and the causal graph structure among latent causal variables. Unlike previous works that necessitate specific graph structures, we do not impose any restrictions on the graph structures among latent causal variables $\mathbf{z}$ other than acyclicity. In addition, we introduce a surrogate variable $\mathbf{u}$ to characterize distribution shifts by modeling the changes in the distribution of $\mathbf{n}$, as well as the causal influences among latent causal variables $\mathbf{z}$. Here $\mathbf{u}$ could be thought of as environment index. More specifically, we parameterize the causal generative models by assuming $\mathbf{n}$ follows an exponential

family given $\mathbf{u}$, and assuming $\mathbf{z}$ and $\mathbf{x}$ are generated as follows:

$$p_{(\mathbf{T},\boldsymbol{\eta})}(\mathbf{n}|\mathbf{u}) := \prod_i \frac{1}{Z_i(\mathbf{u})} \exp[\sum_j \left(T_{i,j}(n_i)\eta_{i,j}(\mathbf{u})\right)], \tag{1}$$

$$z_i := \mathrm{g}_i^{\mathbf{u}}(\mathrm{pa}_i) + n_i, \tag{2}$$

$$\mathbf{x} := \mathbf{f}(\mathbf{z}, \boldsymbol{\varepsilon}), \tag{3}$$

where

- in Eq. 1, $Z_i(\mathbf{u})$ denotes the normalizing constant, and $T_{i,j}(n_i)$ denotes the sufficient statistic for $n_i$, whose natural parameter $\eta_{i,j}(\mathbf{u})$ depends on $\mathbf{u}$. Here we focus on two-parameter (e.g., $j \in \{1, 2\}$) exponential family members, e.g., Gaussian, inverse Gaussian, Gamma, inverse Gamma, and beta distributions as special cases.

- In Eq. 2, the term $\mathrm{pa}_i$ represents the set of parents of $z_i$. $\mathrm{g}_i^{\mathbf{u}}$ signifies a mapping, which can take on various forms, including both linear and nonlinear mappings, and is dependent on $\mathbf{u}$. In addition, there exist common Directed Acyclic Graphs (DAG) constraints among latent causal variables $\mathbf{z}$.

- In Eq. 3, $\mathbf{f}$ denote a nonlinear mapping from $\mathbf{z}$ to $\mathbf{x}$, $\mathbf{x} \in \mathbb{R}^d$ and $\boldsymbol{\varepsilon}$ is independent noise with probability density function $p_{\boldsymbol{\varepsilon}}(\boldsymbol{\varepsilon})$, $\boldsymbol{\varepsilon} \in \mathbb{R}^{d-\ell}$.

The surrogate variable $\mathbf{u}$ plays a crucial role in capturing the distribution shifts in the observed data $\mathbf{x}$. Depending on the task, $\mathbf{u}$ can represent different aspects: environmental indices in domain adaptation or generalization, time indices in time series forecasting Mudelsee (2019), geographic locations (e.g., longitude and latitude) in remote sensing Rußwurm et al. (2020), modality indices in multi-modality datasets, or labels in natural images. With $\mathbf{u}$, distribution shifts could be originated from two main sources: (1) changes in the distributions of the exogenous variables $\mathbf{n}$, modulated by $\mathbf{u}$ as described in Eq. 1, and (2) the causal influences from the parent nodes on each latent causal variable, e.g., $\mathrm{g}_i^{\mathbf{u}}(\mathrm{pa}_i, \mathbf{u})$, also modulated by $\mathbf{u}$ as outlined in Eq. 2. By explicitly modeling these factors, we gain a deeper understanding of how variations in the environment (e.g., $\mathbf{u}$) generate different observed data distributions, which will be further explored in the following sections.

### 3.2 COMPLETE IDENTIFIABILITY RESULTS

Intuitively, distribution shifts—whether caused by environmental changes, system disruptions, or dynamic processes—can be seen as natural interventions on hidden causal variables. These shifts show how these hidden variables influence the observed data in different situations, giving us useful information, such as what changes and what stays the same. When these changes are 'sufficient,' meaning they provide enough information to break symmetries and dependencies that might otherwise obscure causal relationships, we may achieve identifiability. As a result, distribution shifts become a powerful tool for identifying latent causal models, particularly in complex, real-world applications. Specifically, we demonstrate the following results.

**Theorem 3.1.** *Suppose latent causal variables $\mathbf{z}$ and the observed variable $\mathbf{x}$ follow the causal generative models defined in Eqs. 1 - 3. Assume the following holds:*

*(i) The noise probability density function $p_{\boldsymbol{\varepsilon}}(\boldsymbol{\varepsilon})$ must not depend on $\mathbf{u}$ and is always finite,*

*(ii) The function $\mathbf{f}$ in Eq. 3 is smooth and invertible,*

*(iii) There exist $2\ell + 1$ values of $\mathbf{u}$, i.e., $\mathbf{u}_0, \mathbf{u}_1, ..., \mathbf{u}_{2\ell}$, such that the matrix*

$$\mathbf{L} = (\boldsymbol{\eta}(\mathbf{u} = \mathbf{u}_1) - \boldsymbol{\eta}(\mathbf{u} = \mathbf{u}_0), ..., \boldsymbol{\eta}(\mathbf{u} = \mathbf{u}_{2\ell}) - \boldsymbol{\eta}(\mathbf{u} = \mathbf{u}_0)) \tag{4}$$

*of size $2\ell \times 2\ell$ is invertible. Here $\boldsymbol{\eta}(\mathbf{u}) = [\eta_{i,j}(\mathbf{u})]_{i,j}$,*

*(iv) The function class of $\mathrm{g}_i^{\mathbf{u}}$ satisfies the following condition: for each parent node $z_{i'}$ of $z_i$, there exist constants $\mathbf{u}_{i'}$, such that $\frac{\partial \mathrm{g}_i^{\mathbf{u}=\mathbf{u}_{i'}}(\mathrm{pa}_i)}{\partial z_{i'}} = 0$,*

*then the true latent causal variables $\mathbf{z}$ are related to the estimated latent causal variables $\hat{\mathbf{z}}$, which are learned by matching the true marginal data distribution $p(\mathbf{x}|\mathbf{u})$, by the following relationship: $\mathbf{z} = \mathbf{P}\hat{\mathbf{z}} + \mathbf{c}$, where $\mathbf{P}$ denotes the permutation matrix with scaling, $\mathbf{c}$ denotes a constant vector.*

**Proof sketch** The proof can be done according to the following intuition. With the support of assumptions (i)-(iii), we can identify the latent noise variables $\mathbf{n}$ up to permutation and scaling, e.g., $\mathbf{n} = \mathbf{P}\hat{\mathbf{n}} + \mathbf{c}$ where $\hat{\mathbf{n}}$ denotes the recovered latent noise variables obtained by matching the true marginal data distribution. This outcome, in conjunction with the definition in Eq. 2, facilitates the establishment of a mapping between the the true latent causal variables $\mathbf{z}$ and the recovered ones $\hat{\mathbf{z}}$, e.g., $\mathbf{z} = \Phi(\hat{\mathbf{z}})$. Finally, by showing that the Jacobian matrix of $\Phi$ is equivalent to $\mathbf{P}$ if condition (iv) is satisfied, we can conclude the proof. Details can be found in Appendix B.

Assumptions (i)-(iii) are orignally deveoploped by nonlinear ICA (Hyvarinen & Morioka, 2016; Hyvarinen et al., 2019; Khemakhem et al., 2020; Sorrenson et al., 2020). We here ~~consider~~ unitize these assumptions considering the following two main reasons. 1) These assumptions have been verified to be practicable in diverse real-world application scenarios (Kong et al., 2022; Xie et al., 2022b; Wang et al., 2022). 2) Our result eliminates the need to make assumptions about the dimensionality of latent causal or noise variables, ~~which is in contrast to existing methods that require prior knowledge of the dimensionality~~, due to imposing the two-parameter exponential family members on latent noise variables (Sorrenson et al., 2020)[2].

Assumption (iv), originally introduced by this work, is to offer the condition, which characterizes the types of distribution shifts within the context of general latent additive noise models, contributing to identifiability. Assumption (iv), for instance, could arise in the analysis of cell imaging data (e.g., $\mathbf{x}$), where various batches of cells are exposed to different small-molecule compounds (e.g., $\mathbf{u}$). each latent variable (e.g., $z_i$) represents the concentration level of a distinct group of proteins, with protein-protein interactions (e.g., causal relations among $z_i$) playing a significant role (Chandrasekaran et al., 2021). Research has revealed that the mechanisms of action of small molecules exhibit variations in selectivity (Forbes & Krueger, 2019)(Scott et al., 2016), which can profoundly affect protein-protein interactions, e.g., $g_i$. The assumption (iv) requires the existence of a specific $\mathbf{u} = \mathbf{u}_{i'}$, such that the original causal relationship can be disconnected. This parallels cases where small molecule compounds disrupt or inhibit protein-protein interactions (PPIs), effectively causing these interactions to cease (Arkin & Wells, 2004). Such molecules are commonly referred to as inhibitors of PPIs. Developing small molecule inhibitors for PPIs is a key focus in drug discovery (Lu et al., 2020; Bojadzic et al., 2021).

**Remark 3.2** (The Types of Distribution Shifts Contributing to Identifiability). Assumption (iv) is designed to specify the types of distribution shifts that contribute to identifiability, as not all distribution shifts facilitate identifiability. For example, $z_2 := (\lambda'(\mathbf{u}) + b)z_1 + n_2$ is unidentifiable, despite the distribution of of $z_2$ changing across $\mathbf{u}$, whereas $z_2 := (\lambda'(\mathbf{u}))z_1 + n_2$ could be identifiable. To illustrate this concept, consider the following simple example: we can parameterize Eq. 2 as follows: $z_2 := \lambda(\mathbf{u})z_1 + n_2$, where $\lambda(\mathbf{u}) = \lambda'(\mathbf{u}) + b$. As a consequence, while the distribution of $z_2$ shifts across $\mathbf{u}$, ~~there always exists a team~~ $bz_1$ ~~that~~ remains unchanged across $\mathbf{u}$. As a result, the unchanged term $bz_1$ across $\mathbf{u}$ can be absorbed into $\mathbf{f}$ (the mapping from $\mathbf{z}$ to $\mathbf{x}$), resulting in a possible solution $z_2' := \lambda'(\mathbf{u})z_1 + n_2$, not the ~~groundturth~~ groundtruth $z_2 := (\lambda'(\mathbf{u}) + b)z_1 + n_2$, which leads to an unidentifiable outcome. ~~Moreover~~ On the other hand, assumption (iv) implies that we require $\lambda(\mathbf{u} = \mathbf{u}_{i'}) = 0$ and $\lambda'(\mathbf{u} = \mathbf{u}_{i'}) = 0$ (since both ~~$\lambda'$~~ $\lambda$ and $\lambda'$ must belong to the same function class), which results in that $\lambda(\mathbf{u})$ can not be replaced by $\lambda'(\mathbf{u}) + b$ with $b \neq 0$ [3].

**Remark 3.3** (Assumption (iv) does not necessitate the availability of observed data corresponding to the specific $\mathbf{u}_{i'}$). Revisiting the aforementioned example, assumption (iv) is just to limit the function class of $\lambda$, and once samples are drawn from this function class, the assumption is met, allowing observed data corresponding to these samples to be used to infer latent causal variables and their relationships. Therefore, it is not necessary for the sampled data to include the specific point $\mathbf{u}_{i'}$ so that $\lambda(\mathbf{u} = \mathbf{u}_{i'}) = 0$, to generate the corresponding observed data for inference. Importantly, this also highlights the distinction between this work and existing works (Von Kügelgen et al., 2021; Liu et al., 2022; Brehmer et al., 2022; Ahuja et al., 2023; Seigal et al., 2022; Buchholz et al., 2023; Varici et al., 2023), for identifying causal representations. Specifically, existing works typically require that distribution shifts arise from hard interventions to identify causal representations. In contrast, this work proposes that distribution shifts resulting from a function class constrained by Assumption (iv) can also be leveraged for identifiability, which is related to soft interventions. Interestingly,

---

[2]Note that this work employs a special (but not overly restrictive) exponential family to ensure identifiability; for details, refer to Sorrenson et al. (2020)

[3]The statement in this remark holds when $n_i$ are identified up to permutation. A more complicated example can be found in Appendix I.

Assumption (iv) actually requires that the limited function class covers a special point $\mathbf{u}_{i'}$ enabling the removal of incoming edges from parent nodes, which is related to hard intervention and may thus connect to the existing works. Further investigation of Assumption (iv) may provide a bridge between this work and existing works, offering an intriguing direction for future work.

**Remark 3.4** (Latent Causal Graph Structure)**.** Our identifiability result, as established in Theorem 3.1, establishes the identifiability of latent causal variables, thereby implying a unique recovery of the corresponding latent causal graph. This stems from the inherent identifiability of nonlinear additive noise models, as demonstrated in prior research (Hoyer et al., 2008; Peters et al., 2014), irrespective of the scaling applied to $\mathbf{z}$. In addition, linear Gaussian models across multiple environments (e.g., $\mathbf{u}$) are generally identifiable, which is supported by independent causal mechanisms (Huang* et al., 2020; Ghassami et al., 2018; Liu et al., 2022).

### 3.3 Partial Identifiability Results

Condition (iv) in Theorem 3.1, which involves the partial derivatives with respect to each parent node of the variable $z_i$, highlights the requirement for distribution shifts for identifiability. In practice, achieving such distribution shifts for every causal influence from a parent node to $z_i$ may be challenging. In case it is violated, we can still provide partial identifiability results, as follows:

**Theorem 3.5.** *Suppose latent causal variables* $\mathbf{z}$ *and the observed variable* $\mathbf{x}$ *follow the causal generative models defined in Eqs. 1 - 3, and the assumptions (i)-(iii) are satisfied, for each* $z_i$,

(a) *if* ~~it is a root node or~~ *condition (iv) is satisfied, then the true* $z_i$ *is related to the recovered one* $\hat{z}_j$, *obtained by matching the true marginal data distribution* $p(\mathbf{x}|\mathbf{u})$, *by the following relationship:* $z_i = s\hat{z}_j + c$, *where* $s$ *denotes scaling,* $c$ *denotes a constant,*

(b) *if condition (iv) is not satisfied, then* $z_i$ *is unidentifiable.*

**Proof sketch** The proof can be constructed as follows: as mentioned in the proof sketch for Theorem 3.1, with the support of assumptions (i)-(iii), we can establish a mapping between the true latent causal variables $\mathbf{z}$ and the recovered latent causal variables $\hat{\mathbf{z}}$, denoted as $\mathbf{z} = \Phi(\hat{\mathbf{z}})$. By demonstrating that the $i$-th row of the Jacobian matrix of $\Phi$ (corresponding to $z_i$) has one and only one nonzero element when the condition in (a) is met, we can prove (a). Conversely, by showing that if condition (iv) is not satisfied, the $i$-th row of the Jacobian matrix of $\Phi$ (corresponding to $z_i$) has more than one nonzero element, which implies that the true $z_i$ is a composition of more than one recovered variable, we can establish the proof of (b). Details can be found in Appendix C.

**Remark 3.6** (Sufficiency and Necessity of condition (iv))**.** The contrapositive of Theorem 3.5 (b), which asserts that if $z_i$ is identifiable, then condition (iv) is satisfied, serves to establish the necessity of condition (iv) for achieving complete identifiability. This insight, coupled with Theorem 3.1, underscores that condition (iv) is not only sufficient but also necessary for the identifiability result, under assumptions (i)-(iii), without additional assumptions.

**Remark 3.7** (Parent nodes do not impact children)**.** The implications of Theorem 3.5 ((a) and (b)) suggest that $z_i$ remains identifiable, even when its parent nodes are unidentifiable. This is primarily because regardless of whether assumption (iv) is met, assumptions (i)-(iii) ensure that latent noise variables $\mathbf{n}$ can be identified. In the context of additive noise models (or post-nonlinear models discussed in the next section), the mapping from $\mathbf{n}$ to $\mathbf{z}$ is invertible. Therefore, with identifiable noise variables, all necessary information for recovering $\mathbf{z}$ is contained within $\mathbf{n}$. Furthermore, assumption (iv) is actually transformed into relations between each node and the noise of its parent node, as stated in Lemma A.3. As a result, $z_i$ could be identifiable, even when its parent nodes are unidentifiable. *Notably, this partial identifiability property also emphasizes how this work differs from some existing works (Ahuja et al., 2023; Seigal et al., 2022; Buchholz et al., 2023; Varici et al., 2023), which do not provide similar partial identifiability results.*

**Remark 3.8** (Subspace identifiability)**.** The implications of Theorem 3.5 suggest the theoretical possibility of partitioning the entire latent space into two distinct subspaces: latent invariant space containing *invariant* latent causal variables and latent *variant* space comprising variant latent causal variables. This insight could be particularly valuable for applications that prioritize learning invariant latent variables to adapt to changing environments, such as domain adaptation or generalization (Kong et al., 2022). While similar findings have been explored in latent polynomial models in (Liu et al., 2024), this work demonstrates that such results also apply to more flexible additive noise models.

**Summary**    Unlike traditional tasks that emphasize modeling data distributions, causal representation learning seeks to uncover the underlying latent causal mechanisms that generate the observed data. We formalize distribution shifts using the surrogate variable $\mathbf{u}$ within the framework of latent additive noise models, splitting the latent causal mechanisms into two components: one associated with exogenous variables and the other representing causal influences from parent nodes on each latent causal variable. By examining distribution shifts driven by exogenous variables, e.g., assumption (iii) in theorem 3.1, we can identify these latent exogenous variables $\mathbf{n}$ [4]. However, identifying $\mathbf{n}$ does not guarantee component-wise identifiability of $\mathbf{z}$, as demonstrated by Theorem 3.5 (b). By further examining distribution shifts arising from causal influences, e.g., $g_i$ in Eq. 2, assumption (iv) has been proven to be a condition that characterizes the types of distribution shifts for the identifiability of $\mathbf{z}$, supported by Theorem 3.1 and Theorem 3.5 (b). Moreover, we can still achieve partial identifiability when only a subset of $\mathbf{z}$ satisfies assumption (iv), as demonstrated by Theorem 3.5 (a), which may be more practical in real-world applications.

## 4    Extension to Latent Post-Nonlinear Causal Models

While latent additive noise models, as defined in Eq. 2, are general, their capacities are still limited, e.g., requiring additive noise. In this section, we generalize latent additive noise models to latent post-nonlinear models (Zhang & Hyvärinen, 2009), which generally offer more powerful expressive capabilities than latent additive noise models. To this end, we replace Eq. 2 by the following:

$$\bar{z}_i := \bar{g}_i(z_i) = \bar{g}_i(g_i^{\mathbf{u}}(\mathrm{pa}_i) + n_i), \tag{5}$$

where $\bar{g}_i$ denotes a invertible post-nonlinear mapping. It includes the latent additive noise models Eq. 2 as a special case in which the nonlinear distortion $\bar{g}_i$ does not exist. Based on this, we can identify $\bar{\mathbf{z}}$ up to component-wise invertible nonlinear transformation as follows:

**Corollary 4.1.** *Suppose latent causal variables $\mathbf{z}$ and the observed variable $\mathbf{x}$ follow the causal generative models defined in Eqs. 1, 5 and 3. Assume that conditions (i) - (iv) in Theorem 3.1 hold, then the true latent causal variables $\bar{\mathbf{z}}$ are related to the estimated latent causal variables $\hat{\bar{\mathbf{z}}}$, which are learned by matching the true marginal data distribution $p(\mathbf{x}|\mathbf{u})$, by the following relationship: $\bar{\mathbf{z}} = \mathbf{M}_c(\hat{\bar{\mathbf{z}}}) + \mathbf{c}$, where $\mathbf{M}_c$ denotes a component-wise invertible nonlinear mapping with permutation, $\mathbf{c}$ denotes a constant vector.*

**Proof sketch**    The proof can be done intuitively as follows: In Theorem 3.1, the only constraint imposed on the function $\mathbf{f}$ is its injectivity, as mentioned in condition (ii). Therefore, since the function $\bar{g}_i$ is defined as invertible as Eq. 5, we can construct a new injective function $\tilde{\mathbf{f}}$ by composing $\mathbf{f}$ with the function $\bar{\mathbf{g}}$, with each component defined by the function $\bar{g}_i$. This allows us to retain the result derived from Theorem 3.1 and thus conclude the proof. Details can be found in Appendix D.

~~[Latent Causal Graph Structure] Similarly, the identifiability result as established in Corollary 4.1 implies a unique recovery of the corresponding latent causal graph. This stems from the inherent identifiability of nonlinear additive noise models, as demonstrated in prior research (Zhang & Hyvärinen, 2009), irrespective of the component-wise nonlinear scaling applied to $\bar{\mathbf{z}}$. In general, the latent causal graph related to $\bar{\mathbf{z}}$ is the same as one related to $\mathbf{z}$.~~

~~Intuition Due to the assumption that the mapping $\mathbf{f}$, from $\mathbf{z}$ to $\mathbf{x}$, is invertible in latent additive noise models in Eq. 2, the invertible mapping $\bar{g}_i$ in latent post-nonlinear models in Eq. 5 can effectively be incorporated into $\mathbf{f}$. Consequently, the identifiability of latent post-nonlinear models depends on the identifiability of latent additive noise models. This implies that methods specifically designed for latent additive noise models can be directly applied to the recovery of latent post-nonlinear models in the latent space. Furthermore, experimental results obtained from latent additive noise models can also serve as a means to align closely with the identifiability of latent post-nonlinear models, we will discuss in more detail in the experiments.~~

**Model Capacity**    Corollary 4.1 generalizes identifiability of latent additive noise models in Eq. 2 to more complex latent post-nonlinear models. Due to the inherent property of post-nonlinear

---

[4]Note that this is not a straightforward implementation of existing nonlinear ICA. Technically, we must address a gap arising from Eq. 2, to ensure that 1) the mapping from $\mathbf{n}$ to $\mathbf{x}$ is invertible, and 2) the variable $\mathbf{u}$ in Eq. 2 does not compromise the identifiability of $\mathbf{n}$.

models, e.g., invertible component-wise nonlinear mapping $\bar{g}_i$, Corollary 4.1 enables latent additive noise models to uncover causal relationships even when data is generated by latent post-nonlinear models as described in Eq. 5. For example, in cases where data is generated from $z_1 = n_1^3$ and $z_2 = (g_1^{\mathbf{u}}(z_1) + n_2)^3$, despite the presence of multiplicative noise $g_1(z_1)^2 n_2$, Corollary 3.1 supports the effectiveness of latent additive noise models in Eq. 2. Furthermore, in cases where data is generated from $z_1 = n_1^3$ and $z_2 = (\lambda(\mathbf{u})(z_1) + n_2)^3$, although nonlinear relationships are introduced, Corollary 4.1 continues to affirm the applicability of latent linear models in this context.

Similar to Theorem 3.5, we have partial identifiability result as follows:

**Corollary 4.2.** *Suppose latent causal variables $\mathbf{z}$ and the observed variable $\mathbf{x}$ follow the causal generative models defined in Eqs. 1, 5 and 3. Under the condition that the assumptions (i)-(iii) are satisfied, for each $\bar{z}_i$, (a) if it is a root node or condition (iv) is satisfied, then the true $\bar{z}_i$ is related to the recovered one $\hat{\bar{z}}_j$, obtained by matching the true marginal data distribution $p(\mathbf{x}|\mathbf{u})$, by the following relationship: $\bar{z}_i = M_{c,i}(\hat{\bar{z}}_j) + c$, where $M_{c,i}$ denotes a invertible mapping, $c$ denotes a constant, (b) if condition (iv) is not satisfied, then $\bar{z}_i$ is unidentifiable.*

**Proof sketch** The proof can be done intuitively as follows: Again, since the function $\bar{g}_i$ is invertible defined in Eq. 5 and the only constraint imposed the function $\mathbf{f}$ is that $\mathbf{f}$ is invertible in theorem 3.5, we can directly use the result of theorem 3.5 (b) to conclude the proof. Refer to Appendix E.

**Remark 4.3** (Sharing Properties). Corollary 4.2 establishes that the properties outlined in Theorem 3.5, including remark 3.6 to 3.8, remain applicable in latent post-nonlinear causal models.

## 5 LEARNING LATENT ADDITIVE NOISE MODELS

In this section, we translate our theoretical findings into a novel method for learning latent causal models. Our primary focus is on learning additive noise models, as extending the method to latent post-nonlinear models is straightforward, simply involving the utilization of invertible nonlinear mappings as mentioned in **Intuition** for Corollary 4.1. Following previous works in (Liu et al., 2022; 2024), due to permutation indeterminacy in latent space, we can naturally enforce a causal order $z_1 \succ z_2 \succ ..., \succ z_\ell$ without specific semantic information. With guarantee from Theorem 3.1, each variable $z_i$ can be imposed to learn the corresponding latent variables in the correct causal order. As a result, we formulate a prior model as follows:

$$p(\mathbf{z}|\mathbf{u}) = \prod_{i=1}^{\ell} p(z_i|\mathbf{z}_{<i} \odot \mathbf{m}_i, \mathbf{u}) = \prod_{i=1}^{\ell} \mathcal{N}(\mu_{z_i}(\mathbf{z}_{<i} \odot \mathbf{m}_i, \mathbf{u}), \delta_{z_i}^2(\mathbf{z}_{<i} \odot \mathbf{m}_i, \mathbf{u})), \qquad (6)$$

where we focus on latent Gaussian noise variables, considering the re-parametric trick, and we introduce additional vectors $\mathbf{m}_i$, by enforcing sparsity on $\mathbf{m}_i$ and the component-wise product $\odot$, to attentively learn latent causal graph structure. In our implementation, ~~we simply impose $L1$ norm, other methods may also be flexible~~We impose the L1 norm, though other methods may also be flexible, e.g., sparsity priors (Carvalho et al., 2009; Liu et al., 2019). We employ the following variational posterior to approximate the true posterior of $p(\mathbf{z}|\mathbf{x}, \mathbf{u})$:

$$q(\mathbf{z}|\mathbf{u}, \mathbf{x}) = \prod_{i=1}^{\ell} q(z_i|\mathbf{z}_{<i} \odot \mathbf{m}_i, \mathbf{u}, \mathbf{x}), = \prod_{i=1}^{\ell} \mathcal{N}(\mu_{z_i}(\mathbf{z}_{<i} \odot \mathbf{m}_i, \mathbf{u}, \mathbf{x}), \delta_{z_i}^2(\mathbf{z}_{<i} \odot \mathbf{m}_i, \mathbf{u}, \mathbf{x})), \quad (7)$$

where the variational posterior shares the same parameter $\mathbf{m}_i$ to limit both the prior and the variational posterior, maintaining the same latent causal graph structure. Finally, we arrive at the objective:

$$\max \mathbb{E}_{q(\mathbf{z}|\mathbf{x},\mathbf{u})}(p(\mathbf{x}|\mathbf{z}, \mathbf{u})) - D_{KL}(q(\mathbf{z}|\mathbf{x}, \mathbf{u})||p(\mathbf{z}|\mathbf{u})) - \gamma \sum_i \|\mathbf{m}_i\|_1, \qquad (8)$$

where $D_{KL}$ denotes the KL divergence, $\gamma$ denotes a hyperparameters to control the sparsity of latent causal structure. Implementation details can be found in Appendix G.

## 6 EXPERIMENTS

**Synthetic Data** We first conduct experiments on synthetic data, generated by the following process: we divide latent noise variables into $M$ segments, where each segment corresponds

to one value of **u** as the segment label. Within each segment, the location and scale parameters are respectively sampled from uniform priors. After generating latent noise variables, we generate latent causal variables, and finally obtain the observed data samples by an invertible nonlinear mapping on the causal variables. More details can be found in Appendix F.

We evaluate our proposed method (MLPs), implemented by MLPs to model the causal relations among latent causal variables, against established models: vanilla VAE (Kingma & Welling, 2013), $\beta$-VAE (Higgins et al., 2017), identifiable VAE (iVAE) (Khemakhem et al., 2020), and latent polynomial models (Polynomials) (Liu et al., 2024). Notably, the iVAE demonstrates the capability to identify true independent noise variables, subject to certain conditions, with permutation and scaling. Polynomials, while sharing similar assumptions with our proposed method, are prone to certain limitations. Specifically, they may suffer from numerical instability and face challenges due to the exponential growth in the number of terms. While the $\beta$-VAE is popular in disentanglement tasks due to its emphasis on independence among recovered variables, it lacks robust theoretical backing. Our evaluation focuses on two metrics: the Mean of the Pearson

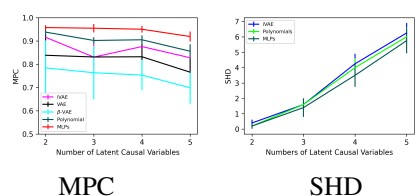

MPC          SHD

Figure 1: In evaluating different methods on latent additive Gaussian noise, we observe distinct performance differences. Notably, the proposed method (MLPs) outperforms others in terms of the MPC, affirming our theoretical results. The right shows the SHD obtained by the proposed method and Polynimals (Liu et al., 2024). Here the estimated graphs of iVAE is obtained by Huang* et al. (2020).

Correlation Coefficient (MPC) to assess performance, and the Structural Hamming Distance (SHD) to gauge the accuracy of the latent causal graphs.

Figure 1 illustrates the comparative performances of various methods, e.g., VAE and iVAE, across different models, e.g., models with different dimensions of latent variables. Based on MPC, the proposed method demonstrates satisfactory results, thereby supporting our identifiability claims. Additionally, Figure 2 presents how the proposed method performs when condition (iv) is not met. It is evident that condition (iv) is a sufficient and necessary condition characterizing the types of distribution shifts for identifiability in the context of latent additive noise models. These empirical findings align with the partial identifiability conclusions discussed in Theorem 3.5.

**Post-Nonlinear Models** In the above experiments, we obtain the observed data samples as derived from a random invertible nonlinear mapping applied to the latent causal variables. The nonlinear mapping can be conceptualized as a combination of an invertible transformation and the specific invertible mapping, $\bar{g}_i$, as mentioned in **Discussion 1** for Corollary 4.1. From this perspective, the results depicted in Figures 1 and 2 also demonstrate the effectiveness of the proposed method in recovering the variables $z_i$ in latent post-nonlinear

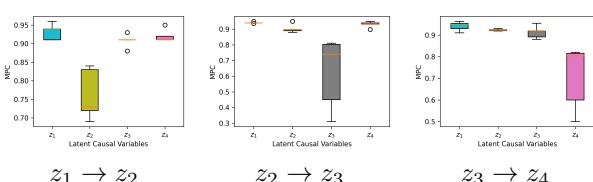

$z_1 \rightarrow z_2$     $z_2 \rightarrow z_3$     $z_3 \rightarrow z_4$

Figure 2: Performance of the proposed method under scenarios where condition (iv) is not satisfied regarding the causal influence of $z_1 \rightarrow z_2$ (consequently, $z_2 \rightarrow z_3$, and $z_3 \rightarrow z_4$). The results are in agreement with partial identifiability in Theorem 3.5.

models Eq. 5, as well as the associated latent causal structures. Consequently, these results also serve to corroborate the assertions in Corollary 4.1 and 4.2, particularly given that $\bar{g}_i$ are invertible.

**Image Data** We further validate our proposed identifiability results and methodology using images from the chemistry dataset introduced by Ke et al. (2021). This dataset is representative of chemical reactions where the state of one element can influence the state of another. The images feature multiple objects with fixed positions, but their colors, representing different states, change according to a predefined causal graph. To align with our theoretical framework, we employ a nonlinear model with additive Gaussian noise for generating latent variables that correspond to the colors of these objects. The established latent causal graph within this context indicates that the 'diamond' object (denoted as $z_1$) influences the 'triangle' ($z_2$), which in turn affects the 'square' ($z_3$). Figure 3

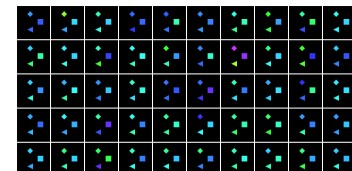

Figure 3: Samples generated by using a modified version of the chemistry dataset originally presented in Ke et al. (2021). In this adaptation, the objects' colors (representing different states) change in accordance with a specified causal graph, e.g., 'diamond' causes 'triangle', and 'triangle' causes 'square'.

provides a visual representation of these observational images, illustrating the causal relationships in a tangible format.

Figure 4 presents MPC outcomes as derived from various methods. Among these, the proposed method demonstrates superior performance. In addition, both the proposed method (MLPs) and Polynomials can accurately learn the causal graph with guarantee. However, Polynomial encounters issues such as numerical instability and exponential growth in terms, which compromises its performance in MPC, as seen in Figure 4. This superiority of MLPs is further evidenced in the intervention results, as depicted in Figure 5. Owing to space constraints, additional traversal results concerning the learned latent variables from other methodologies are detailed in Appendix H. For these methods without identifiability, traversing any learned variable results in a change in color across all objects.

| | $z_1$ | $z_2$ | $z_3$ | | $z_1$ | $z_2$ | $z_3$ | | $z_1$ | $z_2$ | $z_3$ | | $z_1$ | $z_2$ | $z_3$ | | $z_1$ | $z_2$ | $z_3$ |
|---|---|---|---|---|---|---|---|---|---|---|---|---|---|---|---|---|---|---|---|
| $\hat{z}_1$ | 0.089 | 0.094 | 0.857 | $\hat{z}_1$ | 0.067 | 0.582 | 0.628 | $\hat{z}_1$ | 0.095 | 0.631 | 0.683 | $\hat{z}_1$ | 0.862 | 0.281 | 0.003 | $\hat{z}_1$ | 0.912 | 0.501 | 0.024 |
| $\hat{z}_2$ | 0.606 | 0.620 | 0.070 | $\hat{z}_2$ | 0.958 | 0.065 | 0.046 | $\hat{z}_2$ | 0.156 | 0.758 | 0.705 | $\hat{z}_2$ | 0.553 | 0.868 | 0.123 | $\hat{z}_2$ | 0.162 | 0.893 | 0.101 |
| $\hat{z}_3$ | 0.811 | 0.681 | 0.042 | $\hat{z}_3$ | 0.117 | 0.429 | 0.765 | $\hat{z}_3$ | 0.980 | 0.126 | 0.028 | $\hat{z}_3$ | 0.225 | 0.312 | 0.918 | $\hat{z}_3$ | 0.089 | 0.139 | 0.948 |

Figure 4: MPC obtained by different methods on the image dataset. From top to bottom and left to right: VAE, $\beta$-VAE, iVAE, Polynomials, and the proposed method (MLPs). The proposed method performs better than others, which is not only in line with our identifiability claims but also highlights the flexibility of MLPs.

**fMRI Data** Building on the works in (Liu et al., 2022; 2024), we extended the application of the proposed method to the fMRI hippocampus dataset (Laumann & Poldrack, 2015). This dataset comprises signals from six distinct brain regions: perirhinal cortex (PRC), parahippocampal cortex (PHC), entorhinal cortex (ERC), subiculum (Sub), CA1, and CA3/Dentate Gyrus (DG). These signals, recorded during resting states, span 84 consecutive days from a single individual. Each day's data contributes to an 84-dimensional vector, e.g., $\mathbf{u}$. Our focus centers on uncovering latent causal variables, and thus we consider these six brain signals as such, i.e., these signals undergo a random nonlinear mapping to transform them into observable data, then methods can be employed on this transformed data to recover the latent variables.

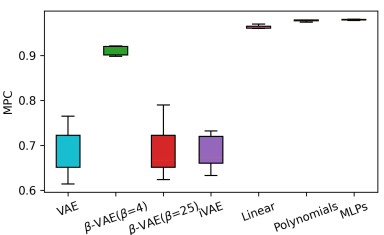

Figure 6: MPC obtained by different methods. Notably, MLPs secure an outstanding average MPC score of 0.981. In comparison, polynomials yield an average MPC score of 0.977, while linear models achieve a slightly lower average MPC score of 0.965.

Figure 6 presents the comparative results yielded by the proposed method alongside various other methods. Notably, the VAE, $\beta$-VAE, and iVAE models presume the independence of latent variables, rendering them incapable of discerning the underlying latent causal structure. Conversely, other methods, including latent linear models, latent polynomials, and latent MLPs, are able to accurately recover the latent causal structure with guarantees. Among these, the MLP models outperform the others in terms of MPC. In the study by Liu et al. (2024), it is noted that linear relationships among the examined signals tend to be more prominent than nonlinear ones. This observation might lead to the presumption that linear models would be effective.

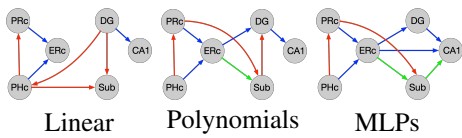

Figure 7: Recovered latent causal structures were analyzed using three distinct approaches: latent linear models, latent polynomials, and latent MLPs. The findings related to latent linear models and latent polynomials are sourced from Liu et al. (2024). Blue edges are feasible given anatomical connectivity, red edges are not, and green edges are reversed.

However, this is not necessarily the case, as these models can still yield suboptimal outcomes. In contrast, MLPs demonstrate superior performance in term of MPC, particularly when compared to polynomial models, which are prone to instability and exponential growth issues. The effectiveness of MLPs is further underscored by their impressive average MPC score of 0.981. This advantage is visually represented in Figure 7, which illustrates the enhanced capability of MLPs.

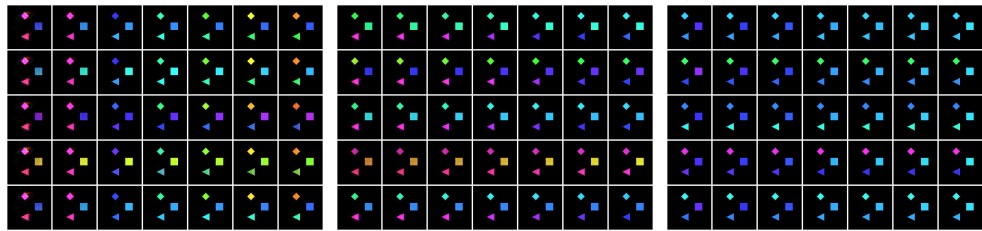

Figure 5: From left to right, the interventions are applied to the causal representations $z_1$, $z_2$, and $z_3$ learned by the proposed method (MLPs), respectively. The vertical axis represents different samples, while the horizontal axis represents the enforcement of various values on the learned causal representation.

## 7 CONCLUSION

This study offers a pivotal contribution by establishing a condition that precisely characterizes the types of distribution shifts for the identifiability of latent additive noise models. Additionally, we present partial identifiability in scenarios where only a subset of distribution shifts fulfills this condition. We then generalize identifiability results to latent post-nonlinear causal models, broadening the scope of its theoretical implications. We translate these theoretical concepts into a practical method, extensive empirical testing was conducted on a diverse array of datasets.

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

# Appendices

## A    Lemmas for the Proposed Latent Causal Models

For ease of proof in the following sections, we first introduce the following lemmas.

**Lemma A.1.** *The mapping between the latent causal variables $\mathbf{z}$ and the recovered latent causal $\hat{\mathbf{z}}$ is independent of $\mathbf{u}$.*

The proof proceeds as follows: According to Eq. 3 and the assumption that the function $\mathbf{f}$ is smooth and invertible (Assumption (ii)), we assume an alternative solution exists such that $\mathbf{x} = \hat{\mathbf{f}}(\hat{\mathbf{z}})$, where $\hat{\mathbf{f}}$ is also invertible. By matching the likelihoods, we obtain $\hat{\mathbf{z}} = \hat{\mathbf{f}}^{-1}(\mathbf{f}(\mathbf{z}, \boldsymbol{\varepsilon}))$. Since $\boldsymbol{\varepsilon}$ is independent of $\mathbf{u}$ (as per Assumption (i)), the proof follows.

**Lemma A.2.** *Denote the mapping from $\mathbf{n}$ to $\mathbf{z}$ as $\mathbf{h}$. This mapping, $\mathbf{h}$, is invertible, and its Jacobian determinant is equal to 1, i.e., $|\det \mathbf{J_h}| = 1$.*

The proof unfolds straightforwardly as follows: Acknowledging that $z_i$ depends contingent on its parents and $n_i$, as delineated in Eq. 2, allows us to iteratively represent $z_i$ in terms of the latent noise variables associated with its parents alongside $n_i$. More explicitly, without loss of the generality, by assuming the true causal order to be $z_1 \succ z_2 \succ ... \succ z_\ell$, we can deduce:

$$
z_1 = \underbrace{n_1}_{h_1(n_1)},
$$
$$
z_2 = \mathrm{g}_2^{\mathbf{u}}(z_1) + n_2 = \underbrace{\mathrm{g}_2^{\mathbf{u}}(n_1) + n_2}_{h_2^{\mathbf{u}}(n_1, n_2)},
$$
$$
z_3 = \underbrace{\mathrm{g}_3^{\mathbf{u}}(n_1, \mathrm{g}_2^{\mathbf{u}}(n_1, \mathbf{u}) + n_2, \mathbf{u}) + n_3}_{h_3^{\mathbf{u}}(n_1, n_2, n_3)}, \tag{9}
$$
$$
......,
$$

where $\mathbf{h^u(n)} = [h_1^{\mathbf{u}}(n_1), h_2^{\mathbf{u}}(n_1, n_2), h_3^{\mathbf{u}}(n_1, n_2, n_3)...]$. Furthermore, according to the additive noise models and DAG constraints, it can be shown that the Jacobi determinant of $\mathbf{h^u}$ equals 1, and thus the mapping $\mathbf{h^u}$ is invertible.

**Lemma A.3.** *Given the assumption (iv) in Theorem 3.1, the partial derivative of $h_i^{\mathbf{u}}(n_1, ..., n_i)$ in Eq. 9 with respect to $n_{i'}$, where $i' < i$, equals 0 when $\mathbf{u}_{i'}$, i.e., $\frac{\partial h_i^{\mathbf{u} = \mathbf{u}_{i'}}(n_1, ..., n_i)}{\partial n_{i'}} = 0$.*

The proof can be constructed as follows: Given that the partial derivative of the mapping $h_i^{\mathbf{u}}(n_1, ..., n_i)$ corresponds to the partial derivative of $\mathrm{g}_i^{\mathbf{u}}$, and leveraging Assumption (iv) in conjunction with the chain rule, we are able to deduce the desired result.

## B  THE PROOF OF THEOREM 3.1

**Theorem 3.1.** *Suppose latent causal variables $\mathbf{z}$ and the observed variable $\mathbf{x}$ follow the causal generative models defined in Eqs. 1 - 3. Assume the following holds:*

*(i) The noise probability density function $p_{\boldsymbol{\varepsilon}}(\boldsymbol{\varepsilon})$ must not depend on $\mathbf{u}$ and is always finite,*

*(ii) The function $\mathbf{f}$ in Eq. 3 is smooth and invertible,*

*(iii) There exist $2\ell + 1$ values of $\mathbf{u}$, i.e., $\mathbf{u}_0, \mathbf{u}_1, ..., \mathbf{u}_{2\ell}$, such that the matrix*

$$\mathbf{L} = (\boldsymbol{\eta}(\mathbf{u} = \mathbf{u}_1) - \boldsymbol{\eta}(\mathbf{u} = \mathbf{u}_0), ..., \boldsymbol{\eta}(\mathbf{u} = \mathbf{u}_{2\ell}) - \boldsymbol{\eta}(\mathbf{u} = \mathbf{u}_0)) \tag{10}$$

*of size $2\ell \times 2\ell$ is invertible. Here $\boldsymbol{\eta}(\mathbf{u}) = [\eta_{i,j}(\mathbf{u})]_{i,j}$,*

*(iv) The function class of $\mathrm{g}_i^{\mathbf{u}}$ satisfies the following condition: for each parent node $z_{i'}$ of $z_i$, there exist constants $\mathbf{u}_{i'}$, such that $\frac{\partial \mathrm{g}_i^{\mathbf{u}=\mathbf{u}_{i'}}(\mathrm{pa}_i)}{\partial z_{i'}} = 0$,*

*then the true latent causal variables $\mathbf{z}$ are related to the estimated latent causal variables $\hat{\mathbf{z}}$, which are learned by matching the true marginal data distribution $p(\mathbf{x}|\mathbf{u})$, by the following relationship: $\mathbf{z} = \mathbf{P}\hat{\mathbf{z}} + \mathbf{c}$, where $\mathbf{P}$ denotes the permutation matrix with scaling, $\mathbf{c}$ denotes a constant vector.*

The proof of Theorem 3.1 unfolds in three distinct steps. Initially, Step I establishes that the identifiability criterion from (Sorrenson et al., 2020) is applicable in our context. Specifically, it confirms that the latent noise variables $\mathbf{n}$ are identifiable, subject only to component-wise scaling and permutation, expressed as $\mathbf{n} = \mathbf{P}\hat{\mathbf{n}} + \mathbf{c}$. Building on this, Step II demonstrates a linkage between the recovered latent causal variables $\hat{\mathbf{z}}$ and the true $\mathbf{z}$, formulated as $\mathbf{z} = \boldsymbol{\Phi}(\hat{\mathbf{z}})$. Finally, Step III utilizes Lemma A.3 to illustrate that the transformation $\boldsymbol{\Phi}$, introduced in Step II, essentially simplifies to a combination of permutation and scaling, articulated as $\mathbf{z} = \mathbf{P}\hat{\mathbf{z}} + \mathbf{c}$.

**Step I:** Suppose we have two sets of parameters $\boldsymbol{\theta} = (\mathbf{f}, \mathbf{T}, \mathbf{h}, \boldsymbol{\eta})$ and $\hat{\boldsymbol{\theta}} = (\hat{\mathbf{f}}, \hat{\mathbf{T}}, \hat{\mathbf{h}}, \hat{\boldsymbol{\eta}})$ corresponding to the same conditional probabilities, i.e., $p_{(\mathbf{f}, \mathbf{T}, \mathbf{h}, \boldsymbol{\eta})}(\mathbf{x}|\mathbf{u}) = p_{(\hat{\mathbf{f}}, \hat{\mathbf{T}}, \hat{\mathbf{h}}, \hat{\boldsymbol{\eta}})}(\mathbf{x}|\mathbf{u})$ for all pairs $(\mathbf{x}, \mathbf{u})$, where $\mathbf{T}$ denote the sufficient statistic of latent noise variables $\mathbf{n}$, and $\mathbf{h}$ is defined in Eq. A.2. Due to the assumption (i), the assumption (ii), and the fact that $\mathbf{h}$ is invertible (e.g., Lemma A.2), by expanding the conditional probabilities via the change of variables formula and taking the logarithm, we have:

$$\log|\det \mathbf{J}_{\mathbf{f}^{-1}}(\mathbf{x})| + \log p_{\boldsymbol{\varepsilon}}(\boldsymbol{\varepsilon}) + \log|\det \mathbf{J}_{\mathbf{h}^{-1}}(\mathbf{z})| + \log p_{(\mathbf{T}, \boldsymbol{\eta})}(\mathbf{n}|\mathbf{u})$$
$$= \log|\det \mathbf{J}_{(\hat{\mathbf{f}} \circ \hat{\mathbf{h}})^{-1}}(\mathbf{x})| + \log p_{(\hat{\mathbf{T}}, \hat{\boldsymbol{\eta}})}(\hat{\mathbf{n}}|\mathbf{u}), \tag{11}$$

where we assume an alternative solution exists such that $\mathbf{x} = \hat{\mathbf{f}}(\hat{\mathbf{z}}) = \hat{\mathbf{f}}(\hat{\mathbf{h}}(\hat{\mathbf{n}}, \mathbf{u}))$. By using the exponential family as defined in Eq. 1, we have:

$$\log|\det \mathbf{J}_{\mathbf{f}^{-1}}(\mathbf{x})| + \log p_{\boldsymbol{\varepsilon}}(\boldsymbol{\varepsilon}) + \log|\det \mathbf{J}_{\mathbf{h}^{-1}}(\mathbf{z})| + \mathbf{T}^T(\mathbf{n})\boldsymbol{\eta}(\mathbf{u}) - \log \prod_i Z_i(\mathbf{u}) = \tag{12}$$

$$\log|\det \mathbf{J}_{(\hat{\mathbf{f}} \circ \hat{\mathbf{h}})^{-1}}(\mathbf{x})| + \hat{\mathbf{T}}^T(\hat{\mathbf{n}})\hat{\boldsymbol{\eta}}(\mathbf{u}) - \log \prod_i \hat{Z}_i(\mathbf{u}), \tag{13}$$

By using Lemma A.2, e.g., $|\det \mathbf{J}_{\mathbf{h}}| = 1$, we have: $|\det \mathbf{J}_{\mathbf{h}^{-1}}| = 1$. Further, since both $\mathbf{h}$ and $\hat{\mathbf{h}}$ must to be the same function class, we also have: $|\det \mathbf{J}_{\hat{\mathbf{h}}^{-1}}| = 1$. Given the above, Eqs. 12-13 can be reduced to:

$$\log|\det \mathbf{J}_{\mathbf{f}^{-1}}(\mathbf{x})| + \log p_{\boldsymbol{\varepsilon}}(\boldsymbol{\varepsilon}) + \mathbf{T}^T(\mathbf{n})\boldsymbol{\eta}(\mathbf{u}) - \log \prod_i Z_i(\mathbf{u}) =$$

$$\log|\det \mathbf{J}_{\hat{\mathbf{f}}^{-1}}(\mathbf{x})| + \hat{\mathbf{T}}^T(\hat{\mathbf{n}})\hat{\boldsymbol{\eta}}(\mathbf{u}) - \log \prod_i \hat{Z}_i(\mathbf{u}). \tag{14}$$

Then by expanding the above at points $\mathbf{u}_l$ and $\mathbf{u}_0$, then using Eq. 14 at point $\mathbf{u}_l$ subtract Eq. 14 at point $\mathbf{u}_0$, we find:

$$\langle \mathbf{T}(\mathbf{n}), \bar{\boldsymbol{\eta}}(\mathbf{u}) \rangle + \sum_i \log \frac{Z_i(\mathbf{u}_0)}{Z_i(\mathbf{u}_l)} = \langle \hat{\mathbf{T}}(\hat{\mathbf{n}}), \bar{\hat{\boldsymbol{\eta}}}(\mathbf{u}) \rangle + \sum_i \log \frac{\hat{Z}_i(\mathbf{u}_0)}{\hat{Z}_i(\mathbf{u}_l)}. \tag{15}$$

Here $\bar{\boldsymbol{\eta}}(\mathbf{u}_l) = \boldsymbol{\eta}(\mathbf{u}_l) - \boldsymbol{\eta}(\mathbf{u}_0)$. By assumption (iii), and combining the $2\ell$ expressions into a single matrix equation, we can write this in terms of $\mathbf{L}$ from assumption (iii),

$$\mathbf{L}^T \mathbf{T}(\mathbf{n}) = \hat{\mathbf{L}}^T \hat{\mathbf{T}}(\hat{\mathbf{n}}) + \mathbf{b}. \tag{16}$$

Since $\mathbf{L}^T$ is invertible, we can multiply this expression by its inverse from the left to get:

$$\mathbf{T}(\mathbf{n}) = \mathbf{A}\hat{\mathbf{T}}(\hat{\mathbf{n}}) + \mathbf{c}, \tag{17}$$

Where $\mathbf{A} = (\mathbf{L}^T)^{-1}\hat{\mathbf{L}}^T$. According to lemma 3 in (Khemakhem et al., 2020) that there exist $k$ distinct values $n_i^1$ to $n_i^k$ such that the derivative $T'(n_i^1), ..., T'(n_i^k)$ are linearly independent, and the fact that each component of $T_{i,j}$ is univariate, we can show that $\mathbf{A}$ is invertible.

Since we assume the noise to be two-parameter exponential family members as defined in Eq. 1, Eq. 17 can be re-expressed as:

$$\begin{pmatrix} \mathbf{T}_1(\mathbf{n}) \\ \mathbf{T}_2(\mathbf{n}) \end{pmatrix} = \mathbf{A} \begin{pmatrix} \hat{\mathbf{T}}_1(\hat{\mathbf{n}}) \\ \hat{\mathbf{T}}_2(\hat{\mathbf{n}}) \end{pmatrix} + \mathbf{c}, \tag{18}$$

Then, we re-express $\mathbf{T}_2$ in term of $\mathbf{T}_1$, e.g., $T_2(n_i) = t(T_1(n_i))$ where $t$ is a nonlinear mapping. As a result, we have from Eq. 18 that: (a) $T_1(n_i)$ can be linear combination of $\hat{\mathbf{T}}_1(\hat{\mathbf{n}})$ and $\hat{\mathbf{T}}_2(\hat{\mathbf{n}})$, and (b) $t(T_1(n_i))$ can also be linear combination of $\hat{\mathbf{T}}_1(\hat{\mathbf{n}})$ and $\hat{\mathbf{T}}_2(\hat{\mathbf{n}})$. This implies the contradiction that both $T_1(n_i)$ and its nonlinear transformation $t(T_1(n_i))$ can be expressed by linear combination of $\hat{\mathbf{T}}_1(\hat{\mathbf{n}})$ and $\hat{\mathbf{T}}_2(\hat{\mathbf{n}})$. This contradiction leads to that $\mathbf{A}$ can be reduced to permutation matrix $\mathbf{P}$ (See APPENDIX C in (Sorrenson et al., 2020) for more details):

$$\mathbf{n} = \mathbf{P}\hat{\mathbf{n}} + \mathbf{c}, \tag{19}$$

where $\mathbf{P}$ denote the permutation matrix with scaling, $\mathbf{c}$ denote a constant vector. Note that this result holds for not only Gaussian, but also inverse Gaussian, Beta, Gamma, and Inverse Gamma (See Table 1 in (Sorrenson et al., 2020)).

**Step II:** By Lemma A.2, we can denote $\mathbf{z}$ and $\hat{\mathbf{z}}$ by:

$$\mathbf{z} = \mathbf{h^u}(\mathbf{n}), \tag{20}$$

$$\hat{\mathbf{z}} = \hat{\mathbf{h}}^{\mathbf{u}}(\hat{\mathbf{n}}), \tag{21}$$

where $\mathbf{h}$ is defined in A.2. Replacing $\mathbf{n}$ and $\hat{\mathbf{n}}$ in Eq. 19 by Eq. 20 and Eq. 21, respectively, we have:

$$(\mathbf{h^u})^{-1}(\mathbf{z}) = \mathbf{P}(\hat{\mathbf{h}}^{\mathbf{u}})^{-1}(\hat{\mathbf{z}}, \mathbf{u}) + \mathbf{c}, \tag{22}$$

where $\mathbf{h}$ (as well as $\hat{\mathbf{h}}$) are invertible supported by Lemma A.2. We can rewrite Eq. 22 as:

$$\mathbf{z} = \mathbf{h^u}(\mathbf{P}(\hat{\mathbf{h}}^{\mathbf{u}})^{-1}(\hat{\mathbf{z}}) + \mathbf{c}). \tag{23}$$

Denote the composition by $\boldsymbol{\Phi}$, we have:

$$\mathbf{z} = \boldsymbol{\Phi}(\hat{\mathbf{z}}). \tag{24}$$

Note that $\boldsymbol{\Phi}$ must also satisfy the condition of being independent of $\mathbf{u}$, as demonstrated by Lemma A.1. Therefore, Consequently, we can remove the dependence on $\mathbf{u}$ in $\boldsymbol{\Phi}$ in Eq. 24.

**Step III** Next, Replacing $\mathbf{z}$ and $\hat{\mathbf{z}}$ in Eq. 24 by Eqs. 19, 20, and 21:

$$\mathbf{h^u}(\mathbf{P}\hat{\mathbf{n}} + \mathbf{c}) = \boldsymbol{\Phi}(\hat{\mathbf{h}}^{\mathbf{u}}(\hat{\mathbf{n}})) \tag{25}$$

By differentiating Eq. 25 with respect to $\hat{\mathbf{n}}$

$$\mathbf{J}_{\mathbf{h^u}}\mathbf{P} = \mathbf{J}_{\boldsymbol{\Phi}}\mathbf{J}_{\hat{\mathbf{h}}^{\mathbf{u}}}. \tag{26}$$

Without loss of generality, let us consider the correct causal order $z_1 \succ z_2 \succ ..., \succ z_\ell$ so that $\mathbf{J}_{\mathbf{h^u}}$ and $\mathbf{J}_{\hat{\mathbf{h}}^{\mathbf{u}}}$ are lower triangular matrices whose the diagonal are 1, and $\mathbf{P}$ is a diagonal matrix with elements $s_{1,1}, s_{2,2}, s_{3,3}, ....$

**Elements above the diagonal of matrix $\mathbf{J_\Phi}$** Since $\mathbf{J_{\hat{h}^u}}$ is a lower triangular matrix, and $\mathbf{P}$ is a diagonal matrix, $\mathbf{J_\Phi}$ must be a lower triangular matrix.

Then by expanding the left side of Eq. 26, we have:

$$
\mathbf{J_{h^u}P} = \begin{pmatrix} s_{1,1} & 0 & 0 & \dots \\ s_{1,1}\frac{\partial h_2^u(n_1,n_2)}{\partial n_1} & s_{2,2} & 0 & \dots \\ s_{1,1}\frac{\partial h_3^u(n_1,n_2,n_3)}{\partial n_1} & s_{2,2}\frac{\partial h_3^u(n_1,n_2,n_3)}{\partial n_2} & s_{3,3} & \dots \\ . & . & . & \dots \end{pmatrix}, \tag{27}
$$

by expanding the right side of Eq. 26, we have:

$$
\mathbf{J_\Phi J_{\hat{h}^u}} = \begin{pmatrix} J_{\Phi_{1,1}} & 0 & 0 & \dots \\ J_{\Phi_{2,1}} + J_{\Phi_{2,2}}\frac{\partial \hat{h}_2^u(n_1,n_2)}{\partial n_1} & J_{\Phi_{2,2}} & 0 & \dots \\ J_{\Phi_{3,1}} + \sum_{i=2}^3 J_{\Phi_{3,i}}\frac{\partial \hat{h}_i^u(n_1,\dots,n_i)}{\partial n_1} & J_{\Phi_{3,2}} + J_{\Phi_{3,3}}\frac{\partial \hat{h}_3^u(n_1,\dots,n_3)}{\partial n_2} & J_{\Phi_{3,3}} & \dots \\ . & . & . & \dots \end{pmatrix}. \tag{28}
$$

**The diagonal of matrix $\mathbf{J_\Phi}$** By comparison between Eq. 27 and Eq. 28, we have $J_{\Phi_{i,i}} = s_{i,i}$

**Elements below the diagonal of matrix $\mathbf{J_\Phi}$** By comparison between Eq. 27 and Eq. 28, and Lemma A.3, for all $i > j$ we have $J_{\Phi_{i,j}} = 0$. For example, ~~G~~given the fact that the equality of two matrices implies element-wise equality, by comparing the corresponding elements of the two matrices Eq. 27 and Eq. 28, e.g., we have $s_{2,2}\frac{\partial h_3^u(n_1,n_2,n_3)}{\partial n_2} = J_{\Phi_{3,2}} + J_{\Phi_{3,3}}\frac{\partial \hat{h}_3^u(n_1,\dots,n_3)}{\partial n_2}$. Then by Lemma A.3, we have a point $\mathbf{u}_{i'}$, so that $\frac{\partial h_3^{\mathbf{u}=\mathbf{u}_{i'}}(n_1,n_2,n_3)}{\partial n_2} = 0$. Further, since both $\mathbf{h^u}$ and $\hat{\mathbf{h}}^u$ must ~~to~~ belong the same function class, we also have: $\frac{\partial \hat{h}_3^{\mathbf{u}=\mathbf{u}_{i'}}(n_1,\dots,n_3)}{\partial n_2} = 0$. Note that $\mathbf{\Phi}$ is independent of $\mathbf{u}$ as mentioned later in Eq. 24, demonstrated by Lemma A.1. As a result, we can use the specific point $\mathbf{u}_{i'}$ to infer $J_{\Phi_{3,2}}$. That is, $J_{\Phi_{3,2}}$ must be 0 across $\mathbf{u}$. Clearly, this result can be extended to the remaining elements $J_{\Phi_{i,j}}$ where $i > j$.

As a result, the matrix $\mathbf{J_\Phi}$ in Eq. 26 equals to the permutation matrix $\mathbf{P}$, which implies that the transformation Eq. 24 reduces to a permutation transformation,

$$
\mathbf{z} = \mathbf{P}\hat{\mathbf{z}} + \mathbf{c}'. \tag{29}
$$

In the preceding proof, it becomes evident that assumption (iv) (or Lemma A.3) is sufficient to constrain the elements below the diagonal of the matrix $\mathbf{J_\Phi}$ to zero. Therefore, our primary objective now shifts to the verification of what happens when assumption (iv) is not met – specifically, whether the claim that the elements below the diagonal of $\mathbf{J_\Phi}$ are zero still holds or not. We will proof that in next section.

## C  THE PROOF OF THEOREM 3.5

**Theorem 3.5.** *Suppose latent causal variables $\mathbf{z}$ and the observed variable $\mathbf{x}$ follow the causal generative models defined in Eqs. 1 - 3, under the condition that the assumptions (i)-(iii) are satisfied, for each $z_i$,*

    *(a) if it is a root node or condition (iv) is satisfied, then the true $z_i$ is related to the recovered one $\hat{z}_j$, obtained by matching the true marginal data distribution $p(\mathbf{x}|\mathbf{u})$, by the following relationship: $z_i = s\hat{z}_j + c$, where $s$ denotes scaling, $c$ denotes a constant,*

    *(b) if condition (iv) is not satisfied, then $z_i$ is unidentifiable.*

Since the proof process in Steps I and II in Appendix B do not depend on the assumption (iv), the results in both Eq. 27 and Eq. 28 hold. Then consider the following two cases.

- In cases where $z_i$ represents a root node or assumption (iv) holds true for $z_i$, by using Lemma A.3, i.e., $\frac{\partial h_i^{\mathbf{u}=\mathbf{u}_{i'}}(n_1,\ldots,n_i)}{\partial n_{i'}} = 0$ and $\frac{\partial \hat{h}_i^{\mathbf{u}=\mathbf{u}_{i'}}(n_1,\ldots,n_i)}{\partial n_{i'}} = 0$ for all $i' < i$, and by comparison between Eq. 27 and Eq. 28, we have: for all $i > j$ we have $J_{\boldsymbol{\Phi}_{i,j}} = 0$, which implies that we can obtain that $z_i = A_{i,i}\hat{z}_i + c_i'$.

- In cases where assumption (iv) does not hold for $z_i$, such as when we compare Eq. 27 with Eq. 28, we are unable to conclude that the $i$-th row of the Jacobian matrix $\mathbf{J}_{\boldsymbol{\Phi}}$ contains only one element. For example, consider $i = 2$, and by comparing Eq. 27 with Eq. 28, we can derive the following equation: $s_{1,1}\frac{\partial h_2^{\mathbf{u}}(n_1,n_2)}{\partial n_1} = J_{\boldsymbol{\Phi}_{2,1}} + J_{\boldsymbol{\Phi}_{2,2}}\frac{\partial \hat{h}_2^{\mathbf{u}}(n_1,n_2)}{\partial n_1}$. In this case, if assumption (iv) does not hold for $z_2$, i.e., there does not exist a point or value $\mathbf{u}_{i'}$ for $\mathbf{u}$ that $\frac{\partial h_2^{\mathbf{u}}(n_1,n_2)}{\partial n_1} = 0$ and $\frac{\partial h_2^{\mathbf{u}}(n_1,n_2)}{\partial n_1} = 0$, then when $J_{\boldsymbol{\Phi}_{2,1}} = s_{1,1}\frac{\partial h_2^{\mathbf{u}}(n_1,n_2)}{\partial n_1} - J_{\boldsymbol{\Phi}_{2,2}}\frac{\partial \hat{h}_2^{\mathbf{u}}(n_1,n_2)}{\partial n_1}$ holds true, we can match the true marginal data distribution $p(\mathbf{x}|\mathbf{u})$. This implies that $J_{\boldsymbol{\Phi}_{2,1}}$ can have a non-zero value. Consequently, $z_2$ can be represented as a combination of $\hat{z}_1$ and $\hat{z}_2$, resulting in unidentifiability. Note that this unidentifiability result also show that the necessity of condition (iv) for achieving complete identifiability, by the contrapositive, i.e., if $z_i$ is identifiable, then condition (iv) is satisfied.

## D  THE PROOF OF COROLLARY 4.1

**Corollary 4.1.** *Suppose latent causal variables $\mathbf{z}$ and the observed variable $\mathbf{x}$ follow the causal generative models defined in Eqs. 1, 5 and 3. Assume that conditions (i) - (iv) in Theorem 3.1 hold, then the true latent causal variables $\bar{\mathbf{z}}$ are related to the estimated latent causal variables $\hat{\bar{\mathbf{z}}}$, which are learned by matching the true marginal data distribution $p(\mathbf{x}|\mathbf{u})$, by the following relationship: $\bar{\mathbf{z}} = \mathbf{M}_c(\hat{\bar{\mathbf{z}}}) + \mathbf{c}$, where $\mathbf{M}_c$ denotes a component-wise invertible nonlinear mapping with permutation, $\mathbf{c}$ denotes a constant vector.*

The proof can be done from the following: since in Theorem 3.1, the only constraint imposed on the function $\mathbf{f}$ is that the function $\mathbf{f}$ is invertible, as mentioned in condition (ii). Consequently, we can create a new function $\widetilde{\mathbf{f}}$ by composing $\mathbf{f}$ with function $\bar{\mathbf{g}}$, in which each component is defined by the function $\bar{g}_i$. Since $\bar{g}_i$ in invertible as defined in Eq. 5, $\widetilde{\mathbf{f}}$ remains invertible. As a result, we can utilize the proof from Appendix B to obtain that $\mathbf{z}$ can be identified up to permutation and scaling, i.e., Eq. 29 holds. Finally, given the existence of a component-wise invertible nonlinear mapping between $\bar{\mathbf{z}}$ and $\mathbf{z}$ as defined in Eq. 5, i.e.,

$$\bar{\mathbf{z}} = \bar{\mathbf{g}}(\mathbf{z}). \tag{30}$$

we can also obtain estimated $\hat{\bar{\mathbf{z}}}$ by enforcing a component-wise invertible nonlinear mapping on the recovered $\hat{\mathbf{z}}$

$$\hat{\bar{\mathbf{z}}} = \hat{\bar{\mathbf{g}}}(\hat{\mathbf{z}}). \tag{31}$$

Replacing $\mathbf{z}$ and $\hat{\mathbf{z}}$ in Eq. 29 by Eq. 30 and Eq. 31, respectively, we have

$$\bar{\mathbf{g}}^{-1}(\bar{\mathbf{z}}) = \mathbf{P}\hat{\bar{\mathbf{g}}}^{-1}(\hat{\bar{\mathbf{z}}}) + \mathbf{c}'. \tag{32}$$

As a result, we conclude the proof.

## E   THE PROOF OF COROLLARY 4.2

**Corollary 4.3.** *Suppose latent causal variables* $\mathbf{z}$ *and the observed variable* $\mathbf{x}$ *follow the causal generative models defined in Eqs. 1, 5 and 3. Under the condition that the assumptions (i)-(iii) are satisfied, for each* $\bar{z}_i$, *(a) if it is a root node or condition (iv) is satisfied, then the true* $\bar{z}_i$ *is related to the recovered one* $\hat{\bar{z}}_j$, *obtained by matching the true marginal data distribution* $p(\mathbf{x}|\mathbf{u})$, *by the following relationship:* $\bar{z}_i = M_{c,i}(\hat{\bar{z}}_j) + c$, *where* $M_{c,i}$ *denotes a invertible mapping,* $c$ *denotes a constant, (b) if condition (iv) is not satisfied, then* $\bar{z}_i$ *is unidentifiable.*

Again, since in Theorem 3.1, the only constraint imposed on the function $\mathbf{f}$ is that the function $\mathbf{f}$ is invertible, as mentioned in condition (ii). Consequently, we can create a new function $\widetilde{\mathbf{f}}$ by composing $\mathbf{f}$ with function $\bar{\mathbf{g}}$, in which each component is defined by the function $\bar{g}_i$. Since $\bar{g}_i$ is invertible as defined in Eq. 5, $\widetilde{\mathbf{f}}$ remains invertible. Given the above, the results in both Eq. 27 and Eq. 28 hold. Then consider the following two cases.

- In cases where $z_i$ represents a root node or assumption (iv) holds true for $z_i$, using the proof in Appendix E we can obtain that $z_i = A_{i,i}\hat{z}_i + c'_i$. Then, given the existence of a component-wise invertible nonlinear mapping between $\bar{z}_i$ and $z_i$ as defined in Eq. 5, we can proof that there is a invertible mapping between the recovered $\hat{\bar{z}}_i$ and the true $\bar{z}_i$.

- In cases where assumption (iv) does not hold for $z_i$, using the proof in Appendix E $z_i$ is unidentifiable, we can directly conclude that $\bar{z}_i$ is also unidentifiable.

## F  Data Details

**Synthetic Data**   In our experimental results using synthetic data, we utilize 50 segments, with each segment containing a sample size of 1000. Furthermore, we explore latent causal or noise variables with dimensions of 2, 3, 4, and 5, respectively. Specifically, our analysis centers around the following structural causal model:

$$n_i :\sim \mathcal{N}(\alpha, \beta), \tag{33}$$
$$z_1 := n_1, \tag{34}$$
$$z_2 := \lambda_{1,2}(\mathbf{u})\sin(z_1) + n_2, \tag{35}$$
$$z_3 := \lambda_{2,3}(\mathbf{u})\cos(z_2) + n_3, \tag{36}$$
$$z_4 := \lambda_{3,4}(\mathbf{u})\log(z_3^2) + n_4, \tag{37}$$
$$z_5 := \lambda_{3,5}(\mathbf{u})\exp(\sin(z_3^2)) + n_5. \tag{38}$$
$$\tag{39}$$

In this context, both $\alpha$ and $\beta$ for Gaussian noise are drawn from uniform distributions within the ranges of $[-2.0, 2.0]$ and $[0.1, 3.0]$, respectively. The values of $\lambda_{i,j}(\mathbf{u})$ are sampled from a uniform distribution spanning $[-2.0, -0.1] \cup [0.1, 2.0]$. After sampling the latent variables, we use a random three-layer feedforward neural network as the mixing function, as described in (Hyvarinen & Morioka, 2016; Hyvarinen et al., 2019; Khemakhem et al., 2020).

**Synthetic Data for Partial Identifiability**   In our experimental results, which utilized synthetic data to explore partial identifiability, we modified the Eqs 33-33 by

$$\dot{z}_i := z_i + z_{i-1}. \tag{40}$$

In this formulation, $\dot{z}_i$ replaces $z_i$. Consequently, for each $i$, there exists a $z_{i-1}$ that remains unaffected by $\mathbf{u}$, thereby violating condition (iv).

**Image Data**   In our experimental results using image data, we consider the following latent structural causal model:

$$n_i :\sim \mathcal{N}(\alpha, \beta), \tag{41}$$
$$z_1 := n_1 \tag{42}$$
$$z_2 := \lambda_{1,2}(\mathbf{u})(\sin(z_1) + z_1) + n_2, \tag{43}$$
$$z_3 := \lambda_{2,3}(\mathbf{u} + y)(\cos(z_2) + z_2) + n_3, \tag{44}$$
$$\tag{45}$$

where both $\alpha$ and $\beta$ for Gaussian noise are drawn from uniform distributions within the ranges of $[-2.0, 2.0]$ and $[0.1, 3.0]$, respectively. The values of $\lambda_{i,j}(\mathbf{u})$ are sampled from a uniform distribution spanning $[-2.0, -0.1] \cup [0.1, 2.0]$.

## G  Implementation Framework

We perform all experiments using the GPU RTX 4090, equipped with 32 GB of memory. Figure 8 illustrates our proposed method for learning latent nonlinear models with additive Gaussian noise. In our experiments with synthetic and fMRI data, we implemented the encoder, decoder, and MLPs using three-layer fully connected networks, complemented by Leaky-ReLU activation functions. For optimization, the Adam optimizer was employed with a learning rate of 0.001. In the case of image data experiments, the prior model also utilized a three-layer fully connected network with Leaky-ReLU activation functions. The encoder and decoder designs were adopted from Liu et al. (2024) and are detailed in Table 1 and Table 2, respectively.

| Input |
|---|
| Leaky-ReLU(Conv2d(3, 32, 4, stride=2, padding=1)) |
| Leaky-ReLU(Conv2d(32, 32, 4, stride=2, padding=1)) |
| Leaky-ReLU(Conv2d(32, 32, 4, stride=2, padding=1)) |
| Leaky-ReLU(Conv2d(32, 32, 4, stride=2, padding=1)) |
| Leaky-ReLU(Linear(32×32×4 + size($\mathbf{u}$), 30)) |
| Leaky-ReLU(Linear(30, 30)) |
| Linear(30, 3*2) |

Table 1: Encoder for the image data.

| Imput |
|---|
| Leaky-ReLU(Linear(3, 30)) |
| Leaky-ReLU(Linear(30, 30)) |
| Leaky-ReLU(Linear(30, 32 × 32 ×4)) |
| Leaky-ReLU(ConvTranspose2d(32, 32, 4, stride=2, padding=1)) |
| Leaky-ReLU(ConvTranspose2d(32, 32, 4, stride=2, padding=1)) |
| Leaky-ReLU(ConvTranspose2d(32, 32, 4, stride=2, padding=1)) |
| ConvTranspose2d(32, 3, 4, stride=2, padding=1) |

Table 2: Decoder for the image data.

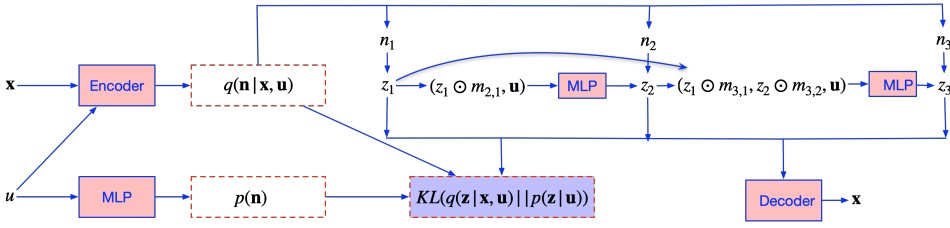

Figure 8: Implementation Framework to learn latnt nonlinear models with non-Gaussian noise. In this example, we demonstrate the method using 3 latent variables, however, our approach is versatile and can be effectively generalized to accommodate much larger graphs.

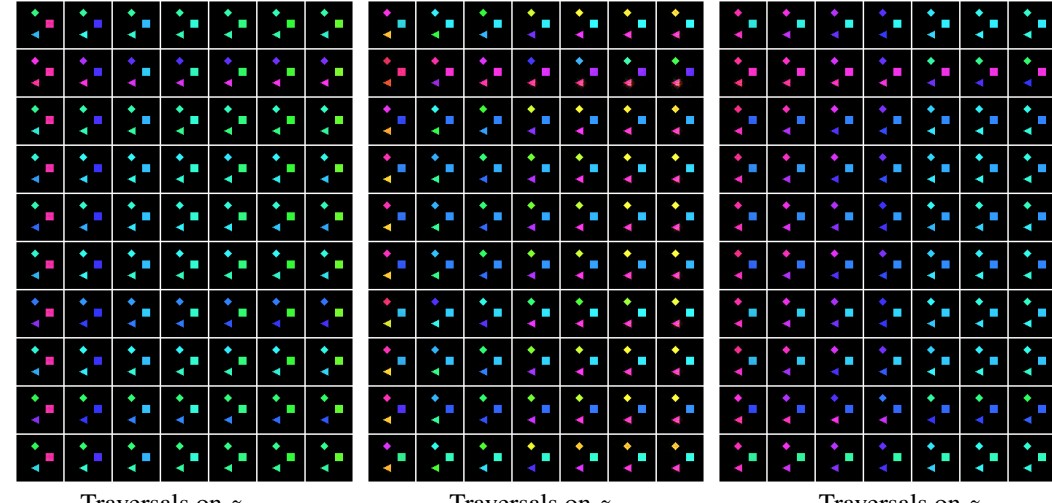

| Traversals on $z_1$ | Traversals on $z_2$ | Traversals on $z_3$ |

Figure 9: The traversal results achieved using VAE on image datasets are depicted. On this representation, the vertical axis corresponds to different data samples, while the horizontal axis illustrates the impact of varying values on the identified causal representation. According to the latent causal graph's ground truth, the 'diamond' variable (denoted as $z_1$) influences the 'triangle' variable ($z_2$), which in turn affects the 'square' variable ($z_3$). Notably, modifications in each of the learned variables lead to observable changes in the color of all depicted objects.

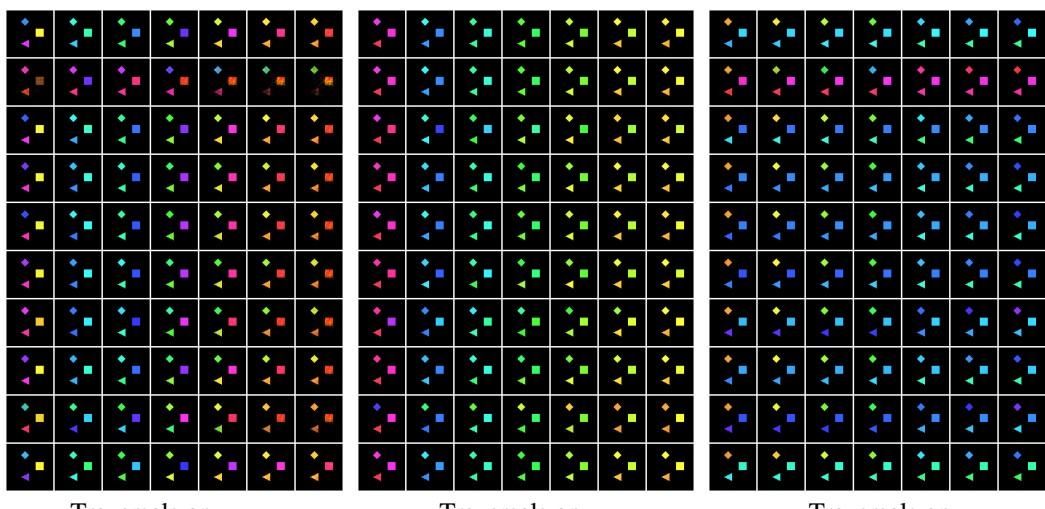

| Traversals on $z_1$ | Traversals on $z_2$ | Traversals on $z_3$ |

Figure 10: The traversal results achieved using $\beta$-VAE on image datasets are depicted. On this representation, the vertical axis corresponds to different data samples, while the horizontal axis illustrates the impact of varying values on the identified causal representation. According to the latent causal graph's ground truth, the 'diamond' variable (denoted as $z_1$) influences the 'triangle' variable ($z_2$), which in turn affects the 'square' variable ($z_3$). Notably, modifications in each of the learned variables lead to observable changes in the color of all depicted objects.

## H    TRAVERSALS ON THE LEARNED VARIABLES BY VAE, $\beta$-VAE, IVAE AND LATENT POLYNOMIALS

## I    COMPARISON OF CONDITION (IV) IN LIU ET AL. (2024) AND THIS WORK

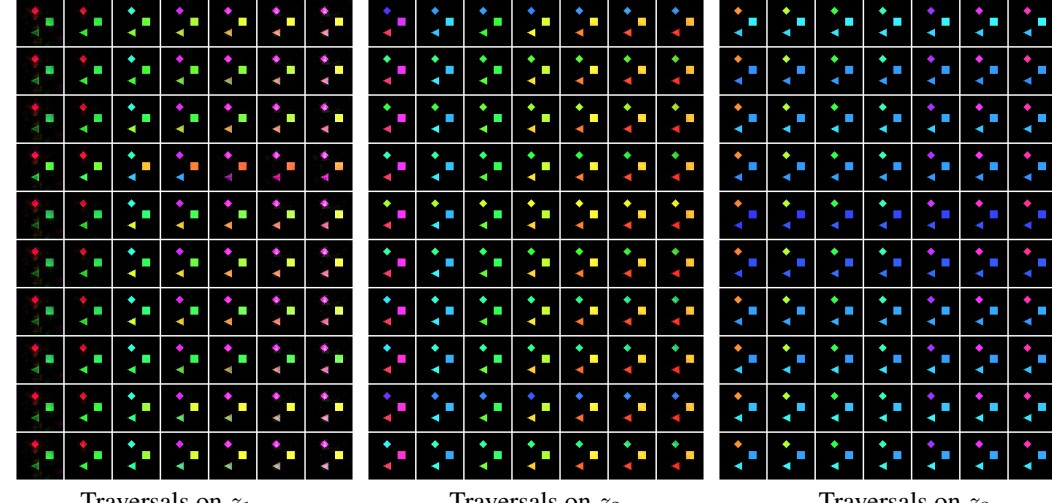

Traversals on $z_1$      Traversals on $z_2$      Traversals on $z_3$

Figure 11: The traversal results achieved using iVAE on image datasets are depicted. On this representation, the vertical axis corresponds to different data samples, while the horizontal axis illustrates the impact of varying values on the identified causal representation. According to the latent causal graph's ground truth, the 'diamond' variable (denoted as $z_1$) influences the 'triangle' variable ($z_2$), which in turn affects the 'square' variable ($z_3$). Notably, modifications in each of the learned variables lead to observable changes in the color of all depicted objects.



Figure 12: From left to right, the interventions are applied to the causal representations $z_1$, $z_2$, and $z_3$ learned by Polynomials, respectively. The vertical axis represents different samples, while the horizontal axis represents the enforcement of various values on the learned causal representation.

In Remark 3.2, we provide a simple example to illustrate condition (iv) in this work. However, this example can also be explained by condition (iv) in the polynomial framework of (Liu et al., 2024), potentially leading to ambiguity in understanding the distinction between condition (iv) in (Liu et al., 2024) and in this work. To clarify these differences and enhance understanding, we present a new example that cannot be captured by the polynomial framework. This example highlights the broader range of distribution shifts contributing to identifiability in our approach. Consider the model $z_2 = \mathrm{g}^{\mathbf{u}}_{2,1}(z_1, \mathbf{u}) + \mathrm{g}^{\mathbf{u}}_{2,2}(z_1) + n_2$, where $\mathrm{g}^{\mathbf{u}}_{2,2}(z_1)$ is not a constant term. In this context, although $\mathrm{g}^{\mathbf{u}}_{2,1}(z_1, \mathbf{u})$ changes across $\mathbf{u}$, leading to shifts in the distribution of $z_2$, the component $\mathrm{g}^{\mathbf{u}}_{2,2}(z_1)$ remains unchanged across different $\mathbf{u}$. This unchanged part, $\mathrm{g}^{\mathbf{u}}_{2,2}(z_1)$ can potentially be absorbed into, resulting in a possible solution $z'_2 = \mathrm{g}^{\mathbf{u}}_{2,1}(z_1, \mathbf{u}) + n_2$, which leads to an unidentifiable outcome. Condition (iv) requires that for the generative model $\mathrm{g}^{\mathbf{u}}_2(z_1, \mathbf{u}) = \mathrm{g}^{\mathbf{u}}_{2,1}(z_1, \mathbf{u}) + \mathrm{g}^{\mathbf{u}}_{2,2}(z_1) + n_2$, so that we have $\frac{\partial \mathrm{g}^{\mathbf{u}}_{2,1}(z_1, \mathbf{u}=\mathbf{u}_{i'})}{\partial z_1} + \frac{\partial \mathrm{g}^{\mathbf{u}}_{2,2}(z_1)}{\partial z_1} = 0$ (a). For the estimated model $z'_2 = \mathrm{g}^{\mathbf{u}}_{2,1}(z_1, \mathbf{u}) + n_2$ to be consistent with the generative model, it must also satisfy condition (iv), $\frac{\partial \mathrm{g}^{\mathbf{u}}_{2,1}(z_1, \mathbf{u}=\mathbf{u}_{i'})}{\partial z_1} = 0$ (b). Comparing (a) and (b), we derive that: $\frac{\partial \mathrm{g}^{\mathbf{u}}_{2,2}(z_1)}{\partial z_1} = 0$ implying that $\mathrm{g}^{\mathbf{u}}_{2,2}(z_1)$ must be a constant term. Consequently, we can exclude the unidentifiable case where $z_2 = \mathrm{g}^{\mathbf{u}}_{2,1}(z_1, \mathbf{u}) + \mathrm{g}^{\mathbf{u}}_{2,2}(z_1) + n_2$, where $\mathrm{g}^{\mathbf{u}}_{2,2}(z_1)$ is not a constant term.

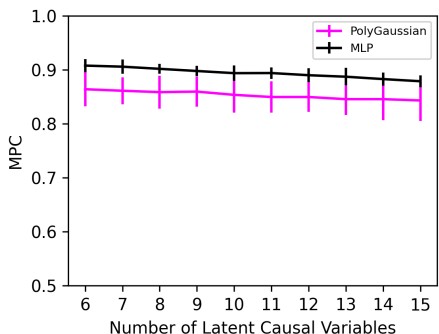

Figure 13: Performances of the proposed method on a large number of latent variables.

## J    MORE RESULTS AND DISCUSSION

In this section, we present additional experimental results on synthetic data to evaluate the effectiveness of the proposed method in scenarios with a large number of latent variables. The performance in these cases is shown in Figure 13. Compared to the polynomial-based approach in Liu et al. (2024), the proposed method, such as MLP, achieves significantly better MCC scores, demonstrating its advantages over polynomials. This superiority becomes particularly evident as the number of latent variables increases. MLPs, being highly flexible, can effectively adapt to the growing complexity. In contrast, when the number of latent variables increases, the number of parent nodes also tends to grow, requiring polynomial-based approaches to incorporate additional nonlinear components to capture the complex relationships among latent variables, which becomes increasingly challenging.

While much of the current work on causal representation learning focuses on foundational identifiability theory, optimization challenges in the latent space remain underexplored. We hope this work not only provides a general theoretical result but also inspires further research on inference methods in the latent space.

## K    HARD INTERVENTION V.S. SOFT INTERVENTION

In general, a hard intervention sets a random variable to a fixed value, effectively removing all incoming edges from its parent nodes in the causal graph and breaking its dependency on original causes. In contrast, a soft intervention modifies or replaces the variable's distribution, typically preserving its incoming edges while altering the distribution. Unlike hard interventions, which completely override a variable's behavior, soft interventions enable more nuanced and flexible modifications to the causal system. More formally, the definition of hard intervention and soft intervention can be found in Massidda et al. (2023). Throughout this paper, references to hard or soft interventions specifically pertain to their application on latent causal variables, such as $z_i$ in Figure 14. To formulate soft intervention, we introduce a surrogate variable $\mathbf{u}$, which acts on the latent causal variables $z_i$ in a causal system as depicted in 14. We use the "red" lines in 14 to represent changes in causal influences among latent causal variables. This differs from the standard definition of edges in causal graphs, which typically indicate causal directions.

## L    COMPARISON WITH MODELS AND METHODS IN LIU ET AL. (2022) AND LIU ET AL. (2024)

Comparison with Generative Models: One key difference between this work and the generative models in Liu et al. (2022) and Liu et al. (2024) is that this work considers additive noise models among latent causal variables, while Liu et al. (2022) assumes linear models and Liu et al. (2024) assumes polynomial models. Additive noise models, as compared to linear models, represent a significant advancement, as they generalize linear models to nonlinear ones. In contrast to polynomials, which have well-known issues such as numerical instability and exponential growth, additive noise models

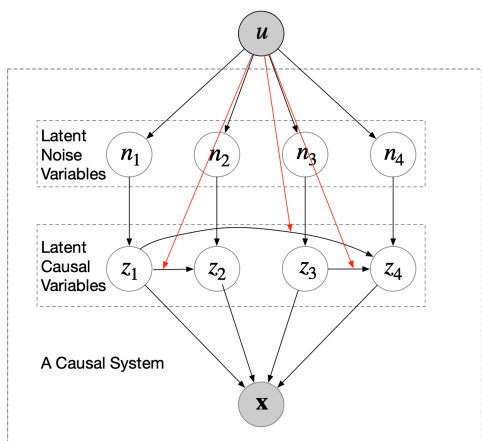

Figure 14: Illustration of a causal system that changes across environments, with a surrogate variable $\mathbf{u}$ is introduced into the causal system to characterize the changing causal mechanisms.

avoid these pitfalls and also facilitate the use of non-parametric models. For instance, the nonlinear component in additive noise can be implemented using flexible network architectures such as MLPs and transformers. This is particularly important, as the success of modern machine learning relies heavily on such complex network architectures.

Comparison with Inference Models: Both this work and previous works in Liu et al. (2022) and Liu et al. (2024) rely on identifiability results from nonlinear ICA. Specifically, nonlinear ICA is used to first identify latent noise variables. As a result, all three methods in the inference process use iVAE, a successful method in nonlinear ICA, to recover latent noise variables. The key difference among these methods lies in how they model the causal influence among latent causal variables, which is due to the differing assumptions on generative models. Specifically, the method in Liu et al. (2022) uses simple linear transformations to infer linear causal influences, while this work uses an MLP to infer causal influence. Compared to the polynomial models in Liu et al. (2024), which are limited by fixed terms for modeling nonlinear relations among latent causal variables e.g., $z_i^n$, additive noise models allow MLPs to model nonlinear relations, offering greater flexibility.

