# OpenReview forum: "Turning Challenges into Opportunities: How Distribution Shifts Enhance Identifiability in Causal Representation Learning"
_ICLR.cc/2025/Conference — Submitted to ICLR 2025_

### Official Review · Reviewer_FhZC · 2024-10-31

**Soundness:** 3
**Presentation:** 2
**Contribution:** 3
**Rating:** 5
**Confidence:** 4

**Summary:**

==Post rebuttal update==

*Soft intervention*. I appreciate the authors' clarification in the rebuttal. The notion itself is clearer, but I still suspect how useful this notion is for understanding the method. Again, I cannot see how previous work's requirement of sample data of specific $u_i$ is related to hard interventions. In the context of current work, incorporating soft intervention seems no more than a notational tweak.

*Assumptions related to the noise distribution*.
- The theory, as in Sorrenson et al. (2020) it is based on, requires certain special two-parameter exponential family members. So, instead of "we focus on two-parameter exponential family members", we should honestly say "we assume..."
- I believe the theory does *not* “eliminate the need to make assumptions about the dimensionality of latent causal or noise variables”, the same as in Sorrenson et al. (2020).

The *“Model Capacity” remark* has not yet been updated.

The following two *similarities to Liu et al. (2022; 2024)* should be mentioned in Related work. 1) They share the same causal graphs (though the current paper does not require $g_i$ to be linear or polynomial). 2) they also use iVAE in the inference process.

**Conditioning on that authors will fix the latter three issues and make a full go-through of the draft to ensure consistent presentation, I am fine with an acceptance. But, given the current draft, I maintain my score 5.**

The issue of soft intervention is harder and I encourage the authors to elaborate more (but a fully satisfiable development could be left for future work.

==End update==

This paper builds on Liu et al. (2022; 2024), which applies nonlinear ICA to noise variables while using ANMs as a bridge between hidden causes and the noises. The main contribution here is relaxing parametric assumptions on causal relationships among hidden causes. Notably, assumption (iv) is shown to be both necessary and sufficient for identifiability, given assumptions (i)-(iii) inherited from nonlinear ICA. Empirical results demonstrate promising performance.

**Strengths:**

The necessary and sufficient condition (iv) is a meaningful theoretical contribution that seems both practically relevant and theoretically significant (though I haven’t verified the proof).

The approach of applying nonlinear ICA to noise variables and ANMs as a bridge is an interesting direction.

**Weaknesses:**

The paper could be more transparent about its limitations and the relationship to prior work; for example, the general exponential family in Eq. (1) and the novelty of the model structure and learning method are not fully clarified.

The concepts of hard vs. soft/natural interventions are unclear and add little to the understanding of the model.

The writing is unclear in places, especially in Remark 3.2 and the “Model Capacity”, which are important because they contain example models.

The experimental comparison with existing causal discovery methods could be added.

**Questions:**

### Hard vs. Soft/Natural Interventions
The notion of “soft intervention” seems to vaguely refer to "intervention different from the standard definition". The only example given is “applying functional transformations,” which are not convincingly “particularly relevant” or “wider”. If we consider Eq. (2), assignments on $u$ could be seen as changing functions applied to $pa_i$, but this is because $u$ and $pa_i$ are separate variables, not because “soft interventions” are inherently broader.

The concept seems unnecessary for the paper, as the assignment on $u$ still formally models distribution shifts, which is a hard intervention, although $u$ may not be intentionally controlled by an agent.

Regarding assumption (iv), I see it does not require data specific to a particular $u_i$ value, but I’m unclear why previous work would. Could you clarify this with an example from previous work? It seems that whether $u$ values are necessary depends on how assumptions are used in proofs (e.g., matrix $L$ is required to construct $P$), which is the thing that needs clarification, not on the hard vs. soft intervention distinction, which seems unnecessary and confusing. Similar comments apply to Remark 3.7 from L291.

If “soft intervention” is central, a formal definition would help, along with connecting it to prior causal inference work (e.g., possibly the natural experiments [1] in economics).

[1] Rosenzweig, M. R., & Wolpin, K. I. (2000). Natural “natural experiments” in economics. Journal of Economic Literature, 38(4), 827-874.

### Transparency
The proposed model extends Liu et al. (2022; 2024) by removing the requirement for $g_i$ to be linear or polynomial, though the paper could clarify that the model itself is not novel.

Learning Method: The proposed approach resembles iVAE with added order and sparsity constraints, which, I believe, were considered in Liu et al. (2022; 2024).

Component-wise identifiability isn’t achieved for general exponential family distributions, as indicated in Sorrenson et al. (2020), which your proof depends on. The two-parameter requirement is for each $z_i$, and $l$ is unrestricted also in prior work. So I think your method does not differ from “existing methods that require prior knowledge of dimensionality due to the two-parameter exponential family”?

### Writing
In Remark 3.2, it would help to clarify that an unidentifiable and identifiable example are being considered from the outset. Presenting the identifiable example first might improve readability. Phrasing such as “Moreover” could be replaced with “On the other hand.” Better examples could be provided; currently, the unidentifiable example violates both (iii) and (iv), while the identifiable example satisfies both. A clearer illustration would be examples that satisfy (iii) but only one satisfies (iv).

In Remark 3.4, the sentence “In addition, linear Gaussian models…” seems unrelated and could be removed (though itself correct).

In Thm. 3.5(a), a root node vacuously satisfies (iv), so it would be clearer to state this outside the theorem and remove “if it is a root node or.”

L307: I think identifying $n$ does guarantee identifiability of $z$, though not component-wise identifiability of each $z_i$.

Both Remark 4.2 and Intuition regarding Corollary 4.1 could be removed, as they merely repeat “we can construct a new injective function f by composing f with …” in the proof sketch. Similarly, the proof sketch for Corollary 4.3 could be omitted.

The “Model Capacity” explanations are confusing, with a confusion between Thm. 3.1 and Cor. 4.1, and the examples are not clearly explained, e.g., the second example assumes $\lambda(u)=0$ for some $u$ values.

### Experiments
It would be informative to see if this method outperforms "iVAE to recover $z$ and then existing causal discovery methods to build causal graphs among $z$", particularly in the cases of Fig. 1 (right) and Fig. 7.

The synthetic and image data are limited to chain structures; if time permits, additional synthetic data with more complex structures would add value.

Why does $\beta$-VAE perform so well on the fMRI data for reasonable $\beta$ values?

### Mino/typo:

L62: “Given space constraints” could be removed.

L66: MLPs are not particularly “advanced architectures.”

L239: Typo, $\lambda’$ appears twice.

L445: “Discussion 1” →  “Intuition.” However, per the “Writing” section, “as mentioned in Discussion 1 for Corollary 4.1” could be omitted.

L483: Typo, “top to bottom and…”

---

> ### Author Response · Authors · 2024-11-23
>
> We thank Reviewer FhZC for taking the time to review our submission and for their positive feedback, including a meaningful theoretical contribution, as a bridge is an interesting direction. Below, we address the reviewer’s concerns on presentation.
>
> ----
>
> **Q1:** The notion of “soft intervention” seems to vaguely refer to "intervention different from the standard definition"...  The concept seems unnecessary for the paper,....If “soft intervention” is central, a formal definition would help..
>
> **R1:** We greatly appreciate the reviewer for pointing out this concern. We fully acknowledge the issue raised, as without a clear definition, confusion can easily arise. To address this, we have added a new Section K in the Appendix. In this section, we clarify that u is a surrogate variable introduced into a causal system to characterize changing causal mechanisms, and emphasize that u should not be interpreted as a variable within the causal system itself. We also highlight the distinction between soft and hard interventions, not only in the general understanding of these terms but also by citing a paper that provides a formal definition. Additionally, we have modified the position of u in Eq. (2) from being an input to a superscript, aligning it more closely with the representation of soft interventions. Furthermore, we clarify that when we refer to interventions, we are generally discussing interventions acting on latent causal variables, e.g., $z_i$. More details on these points can be found in the newly added Section K in the Appendix.
>
> **Q2:** Regarding assumption (iv), I see it does not require data specific to a particular value, but I’m unclear why..... It seems that whether values are necessary depends on how assumptions are used in proofs...
>
> **R2:** Indeed, whether the values of u are necessary depends on how the assumptions are applied in the proofs. We have updated this section for greater clarity, thank you once again for your careful review. Essentially, the necessity of u depends on the specific role the assumptions play in the proof. The logic follows that different proof techniques lead to different conditions for identifiability. In this work, we show that a limited function class, such as the one constrained by assumption (iv), can contribute to identifiability. Interestingly, assumption (iv) requires the existence of a mechanism for removing incoming edges from parent nodes, which is closely related to the concept of hard interventions used for identifiability in existing work. Further exploration of assumption (iv) could provide a new bridge connecting this work with existing studies, presenting an exciting avenue.
>
> **Q3:** Comparison with Liu et al. (2022; 2024) in Generative Models and Inference Methods.
>
> **R3:** We have added a new Section L in the Appendix of the updated version. In this section, we detail the differences between our work and previous studies in generative models, as well as provide a comparison of inference methods. For more details, please refer to Section L.
>
> **Q4:** Component-wise identifiability isn’t achieved for general exponential family distributions....
>
> **R4:** Thank you for your rigorous review. We have revised the relevant parts accordingly and added a footnote to highlight that we use the two parameters from Sørenson et al. (2020) in a limited manner.
>
> **Q5:** In Remark 3.2, it would help to clarify that an unidentifiable and identifiable example are being considered from the outset....A clearer illustration would be examples that satisfy (iii) but only one satisfies (iv).
>
> **R5** We have revised the relevant part in Remark 3.2. Additionally, we have provided another example for stronger understanding in Section I of the Appendix.
>
> **Q6:** In Remark 3.4, ....In Thm. 3.5(a), L307: I think identifying...Both Remark 4.2 and Intuition regarding Corollary 4.1 could be removed,..
>
> **R6:** We have revised these parts accordingly. Thank you for your suggestion!
>
> **Q7:** It would be informative to see if this method outperforms "iVAE to recover and then existing causal discovery methods to build causal graphs among,.... if time permits, additional synthetic data
>
> **R7:** We have added the results from iVAE + CNNOD  [1] as shown in Figure 1. We have also added a new Section J to present results with a larger number of latent causal variables.
>
>
> [1] Huang, Biwei, et al. "Causal discovery from heterogeneous/nonstationary data." Journal of Machine Learning Research 21.89 (2020): 1-53.

---

> > ### Author Response · Authors · 2024-11-23
> >
> > **Q8:** Why does beta-VAE perform so well on the fMRI data for reasonable beta values?
> >
> > **R8** It is difficult to justify why such a result appears, and we suspect it may be due to the specific properties of the fMRI data. On the other hand, beta-VAE suffers from two critical limitations: 1) In practical applications, determining an appropriate value for $\beta$ is challenging due to the inherent uncertainty in the underlying latent variables. 2) As the value of $\beta$ increases, there is an inevitable trade-off between disentanglement and reconstruction quality. Liu et al. (2022) provide a detailed discussion of these limitations and offer experimental results showing how performance changes with different values of $\beta$. For more details, please refer to Liu et al. (2022).

---

> > > ### Author Response · Authors · 2024-11-29
> > > **Follow-Up on Discussion Phase**
> > >
> > > Dear Reviewer FhZC,
> > >
> > > We have revised our draft to address your concerns regarding its presentation, and we sincerely appreciate your valuable feedback.
> > >
> > > As the discussion phase nears its conclusion, we kindly ask if your concerns have been addressed. We understand you may have a busy schedule, but if you have any follow-up questions or remaining issues, please feel free to let us know.
> > >
> > > If our rebuttal has resolved your concerns, we would be grateful if you could update your score and provide your feedback.
> > >
> > > Thank you for your time and effort!
> > >
> > > Best regards,
> > >
> > > Authors

---

> > > > ### Author Response · Authors · 2024-12-02
> > > > **Could you kindly verify if the provided clarification addresses your concerns?**
> > > >
> > > > Dear Reviewer FhZC,
> > > >
> > > > As the discussion phase deadline is approaching, we kindly request your feedback on whether our response has adequately addressed the concerns you raised. Your confirmation would greatly help us finalize our revisions and ensure that all your points have been sufficiently addressed.
> > > >
> > > > Thank you for your time and consideration. We truly appreciate your valuable input.
> > > >
> > > > Best,
> > > >
> > > > Authors

---

> > > > > ### Author Response · Authors · 2024-12-03
> > > > > **Urgent Reminder: Your Feedback is Crucial for Finalizing Our Draft**
> > > > >
> > > > > Dear Reviewer FhZC,
> > > > >
> > > > > As the deadline is approaching, we wanted to kindly remind you of our request for clarification on the concerns raised in your review. Your feedback is incredibly important to us, and we have dedicated significant time and effort to carefully revise the draft in accordance with your suggestions.
> > > > >
> > > > > We fully understand that you are busy, but your response is crucial for the final stages of our work. We would greatly appreciate it if you could provide your insights at your earliest convenience.
> > > > >
> > > > > Thank you once again for your valuable time and support.
> > > > >
> > > > > Best regards,
> > > > >
> > > > > Authors

---

### Official Review · Reviewer_Rg2d · 2024-11-01

**Soundness:** 3
**Presentation:** 3
**Contribution:** 2
**Rating:** 5
**Confidence:** 3

**Summary:**

This paper develops a non-parametric model of latent additive noise models to investigate the identifiability problem of general latent causal variables, extending previous linear [Liu et al., 2022] and polynomial [Liu et al., 2024] models to more general nonlinear cases. Specifically, the core Theorems 3.1 and 3.5 proposed in this work categorize distribution shifts induced by soft interventions, demonstrating that the latent causal variables $z_i$ are identifiable under certain assumptions. The authors further generalize latent additive noise models to latent post-nonlinear models to better accommodate real-world application scenarios. Finally, the learning algorithm developed based on their theoretical framework demonstrates promising performance on extensive synthetic and real-world datasets.

**Strengths:**

- The paper is well-structured and written fluently, with the various assumptions needed for the theoretical results clearly presented.

- The explanations of intuition and insights are particularly excellent, making the paper easy to understand and enhancing its readability.

- The main results of the paper are meaningful and insightful for real-world applications. Theorems on the partial identifiability of latent causal variables provide valuable insights and suggest feasible future approaches in the fields of domain generalization and domain adaptation.

**Weaknesses:**

- The paper relies on [Liu et al., 2024], extending the structural causal equation $g_i$ from polynomial functions to nonlinear functions, and modifying condition (iv) from the existence of a function class $\lambda_{i,j}(\mathbf{u})=0$ to the existence of an environment $\mathbf{u}_{i^\prime}$ such that the partial derivative of $g_i$ with respect to each parent node is zero. Although the relaxation of these assumptions significantly enhances the model's applicability to real-world scenarios, the core motivation of leveraging distribution shifts to assess identifiability based on changes in the coefficients of $z_i$ overlaps substantially with [Liu et al., 2024]. Thus, as a follow-up work, I tend to view this paper as an incremental improvement.
- What is the essential difference between the nonlinear assumption in condition (iv) and the scenario where the coefficient of $z_i$ is zero? Is $\frac{\partial g_i}{\partial z_{i}^{\prime}}=0$ equivalent to having the coefficient $\lambda_{i,j}(\mathbf{u})=0$ in $\lambda_{i,j}(\mathbf{u})h(z_{i}^{\prime})+n$? For example, in the polynomial case $z_2=\lambda_{1,2}(\mathbf{u})z^2_1+n_2$ and the nonlinear case $z_2=\lambda_{1,2}(\mathbf{u})cos(z_1)+n_2$, it seems that condition (iv) can be uniformly reduced to the coefficient $\lambda_{1,2}(\mathbf{u})=0$. I would like to ask what makes the nonlinear condition (iv) particularly different from the scenario where the coefficient $\lambda_{i,j}(\mathbf{u})=0$ in the polynomial case.
- Could you provide some examples of unidentifiable nonlinear $g_i$? I noticed that the examples in Remark 3.2 and the Model Capacity section are all polynomial. It would be helpful if the authors could include examples illustrating condition (iv) in the nonlinear case to aid readers in understanding how it differs from condition (iv) in [Liu et al., 2024].

**Some Minor Issues**:

- Are the number of latent causal variables and the causal order given as prior information? How does the proposed theory perform with a larger number of latent causal variables? I observed that the experiments were conducted with 3-6 latent causal variables.
- More visual examples would help readers better understand these assumptions and their implications, especially indicating in a causal graph which $z_i$ are identifiable and which are not.

**Questions:**

- In the part of Remark 3.8 (Subspace identifiability), are the latent causal variables $z_i$ that vary across domains identifiable? In the context of OOD generalization, does the invariant feature $C$ violate condition (iv), while the spurious feature $S$ satisfies condition (iv)?
- Line 208: definition in 2 -> definition in Eq. 2
- Line 235: team $bz_1$ -> term $bz_1$
- Line 239: $\lambda^{\prime}$ and $\lambda^{\prime}$ -> $\lambda$ and $\lambda^{\prime}$
- In the paragraph "Model Capacity," should Corollary 3.1 on line 356 and line 358 be changed to Corollary 4.1?
- Line 437: Corollary 3.5 -> Theorem 3.5?

---

> ### Author Response · Authors · 2024-11-23
> **Response**
>
> We thank Reviewer Rg2d for taking the time to review our submission and for their positive feedback, including recognition of our work as well-structured and fluently written, with excellent intuition and insights, as well as meaningful and impactful results for real-world applications. Below, we address the reviewer’s concerns in detail.
>
> -----
>
> **Q1:** The paper relies on [Liu et al., 2024], extending ..to nonlinear functions, and modifying condition (iv).. Although the relaxation of these assumptions significantly enhances the model's applicability to real-world scenarios, the core motivation..overlaps substantially with [Liu et al., 2024]. Thus, as a follow-up work, I tend to view this paper as an incremental improvement.
>
> **R1:** Thank you for the detailed comparison!
>
> First, we would like to emphasize that using distribution shifts for identifiability is a well-established research direction, with many existing works as mentioned introduction and related work. The key difference among these works lies in considering which types of distributions apply under which function class.
>
> Second, in additive noise models, the nonlinear component is not solely based on changes in the coefficients. Since the nonlinear part is entirely nonparametric, the changes encompass the entire nonlinear mapping. For example, the changes could involve $\lambda$ in $z_2=\lambda_1 z_1 + \lambda_1 z^2_1 +n_2$, or they could involve $\beta$, $z_2= g(z_1,\beta) +n_2$ where the changes occur within the unknown function $g$, beyond the coefficients.
>
> Third, one significant reason for the success of modern machine learning is its reliance on sophisticated structural units, such as transformers. These complex structures often correspond to intricate function relationships and are typically non-parametric, as seen in models like MLPs. The nonparametric components in additive noise models allow for complex structures that go beyond the limitations of polynomial forms, which are restricted to terms such as quadratic or cubic.
>
> Furthermore, we extend our results to post-nonlinear causal models, which are more powerful than additive noise models, i.e., they accommodate multiplicative noise.
>
> Finally, we emphasize that progressing toward a general result is a long-standing research endeavor, and obtaining such results is typically not a one-step process. In this long process, this work serves as a foundational step, generalizing some previous results and providing a potential cornerstone toward a general result.
>
> **Q2:** What is the essential difference between the nonlinear assumption in condition (iv) and the scenario where the coefficient of is zero?
>
> **R2:** Note that here function $g_i$ is non-parametric. Therefore, condition (iv) is specifically designed for such non-parametric functions to define the boundary of identifiability. Unlike parameter-specific conditions that are restricted to particular cases, such as polynomials with zero coefficients or the nonlinear case $sin(z_1)$ you mentioned, our condition transcends these limitations. It offers a comprehensive and general framework applicable to all function classes, including special cases like polynomials where the coefficient equals zero. This implies that if changes in causal influences satisfy condition (iv), identifiability is guaranteed; otherwise, it is not, irrespective of the underlying function class. The strength of this non-parametric approach lies in its broad applicability and robustness, greatly expanding the scope and practical importance of our identifiability analysis.

---

> > ### Author Response · Authors · 2024-11-23
> >
> > **Q3:** Could you provide some examples of unidentifiable nonlinear? I noticed that the examples in Remark 3.2 and the Model Capacity section are all polynomial. It would be helpful if the authors could include examples illustrating condition (iv) in the nonlinear case to aid readers in understanding how it differs from condition (iv) in [Liu et al., 2024].
> >
> > **R3:** Thank you for your suggestion. To facilitate understanding of condition (iv), we initially provided a simple example in Remark 3.2. Here, we delve into more complex cases for comparison with condition (iv) from [Liu et al., 2024]. For example, consider the model $z_2= g_{2,1}(z_1,u) + g_{2,2}(z_1) + n_2$, where $g_{2,2}(z_1)$ is not a constant term. In this context, although $g_{2,1}(z_1,u)$ changes across u, leading to shifts in the distribution of $z_2$, the component $g_{2,2}(z1)$ remains unchanged across different u. This unchanged part, $g_{2,2}(z _ 1) $ can potentially be absorbed into, resulting in a possible solution $z' _ 2=g _ {2,1}(z _ 1,u) + n _ 2$, which leads to an unidentifiable outcome. Condition (iv) requires that for the generative model $g _ {2}(z _ 1,u) = g _ {2,1}(z _ 1,u) + g _ {2,2}(z _ 1) + n _ 2$, so that we have $\frac{\partial {\mathrm{g}^{\mathbf{u}} _ {2,1}(z _ 1,\mathbf{u}=\mathbf{u} _ {i'})}}{\partial z _ {1}} + \frac{\partial {\mathrm{g}^{\mathbf{u}} _ {2,2}(z _ 1)}}{\partial z _ {1}}=0$ (a). For the estimated model $z' _ 2=g _ {2,1}(z _ 1,u) + n _ 2$ to be consistent with the generative model, it must also satisfy condition (iv),  $\frac{\partial {\mathrm{g}^{\mathbf{u}} _ {2,1}(z _ 1,\mathbf{u}=\mathbf{u} _ {i'})}}{\partial z _ {1}}=0$ (b). Comparing (a) and (b), we derive that: $\frac{\partial {\mathrm{g}^{\mathbf{u}} _ {2,2}(z _ 1)}}{\partial z _ {1}}=0$，implying that $g _ {2,2}(z _ 1)$ must be a constant term. Consequently, we can exclude the unidentifiable case where $z _ 2= g _ {2,1}(z _ 1,u) + g _ {2,2}(z _ 1) + n _ 2$, where $g _ {2,2}(z _ 1)$ is not a constant term. We have added a section in appendix I to make the comparison between condition (iv) in [Liu et al., 2024] and condition (iv) in this work more clear.
> >
> > **Q4:** Are the number of latent causal variables and the causal order given as prior information?
> >
> > **R4:** As we mentioned, our result eliminates the need to make assumptions about the dimensionality of latent causal or noise variables. We do not require the causal order given as prior information. In inference model, we enforce a causal order without specific semantic information, due to permutation indeterminacy in latent space.  This does not mean that we use a causal order prior. To illustrate this concept, consider a scenario with only two latent causal variables, namely size (z1) and color (z2) of an object, with z1 causing z2. Upon obtaining two recovered latent causal variables, z'1 and z'2, permutation indeterminacy arises, allowing z'1 to correspond to either the size (z1) or the color (z2). This flexibility empowers us to pre-define a causal order in the inference model, such as z'1 causing z'2, without specifying semantic information (size or color) for the nodes. Consequently, the inference model establishes a causal relationship between z'1 and z'2, prompting them to learn the genuine latent semantic information. In other words, the inference model compels z'1 (or z'2) to acquire size (or color) information, effectively sidestepping DAG constraints. This strategic use of permutation indeterminacy enhances the adaptability of our approach. For a more comprehensive and detailed explanation, please refer to Section 3.4 of the work by Liu et al. (2022), where the authors provide a thorough explanation elucidating the rationale behind the predefinition of a causal order in inference models.
> >
> > **Q5:** How does the proposed theory perform with a larger number of latent causal variables.
> >
> > **Q5:** For identifiability analysis, we generally assume that global optimization is achievable. Therefore, the proposed theory remains applicable even for a larger number of latent causal variables. Additionally, causal representation learning is still in its early stages, with much of the existing research focusing on key aspects such as identifiability. One of the primary challenges in this field is solving the optimization problem within the latent space. Typically, only approximations of the true latent variables can be obtained, and even small inaccuracies in these approximations can result in significantly different learned causal graphs. This challenge partly explains why many studies are conducted in settings where the latent variable dimensions are relatively low, often below 10. We have added a section in Appendix J to present results for cases with a larger number of latent causal variables. Thank you for your suggestion.

---

> > > ### Author Response · Authors · 2024-11-23
> > >
> > > **Q6:** In the part of Remark 3.8 (Subspace identifiability), are the latent causal variables that vary across domains identifiable? In the context of OOD generalization, does the invariant feature c violate condition (iv), while the spurious feature s satisfies condition (iv)?
> > >
> > >
> > > **R6:** The discussion on subspace identifiability aims to highlight the potential applications of partial identifiability. For out-of-distribution (OOD) generalization, the chosen approach depends heavily on the specific setting, such as covariate shifts or label shifts. Each setting involves different assumptions about changes in the latent context c and style variables s. For instance, in the context of covariate shifts, the latent style s is generally considered to change, whereas in label shift settings, it is typically the latent context c changes. Therefore, whether c or s violates condition (iv) depends on the specific setting. However, regardless of what changes or remains unchanged, this information can be leveraged to identify partial variables by applying our partial identifiability results.
> > >
> > > ____
> > >
> > >
> > > We have also reviewed and corrected typos in the manuscript. Thank you for your detailed feedback!

---

> > > > ### Author Response · Authors · 2024-11-29
> > > > **Follow-Up on Discussion Phase**
> > > >
> > > > Dear Reviewer Rg2d,
> > > >
> > > > Thank you again for your thoughtful feedback. We have carefully incorporated your suggestions into the revised version and deeply value your insights.
> > > >
> > > > As the discussion phase draws to a close, we kindly ask whether the updates address your concerns. If there are any unresolved issues or additional questions, please don’t hesitate to share them with us.
> > > >
> > > > If the revisions meet your expectations, we would greatly appreciate it if you could reflect this in your score and share your closing remarks.
> > > >
> > > > Thank you for your time and consideration!
> > > >
> > > > Best regards,
> > > >
> > > > The Authors

---

> > > > > ### Author Response · Authors · 2024-12-02
> > > > > **Follow-Up on Addressing Your Concerns**
> > > > >
> > > > > Dear Reviewer Rg2d,
> > > > >
> > > > > As the discussion phase deadline approaches, we kindly ask if you could let us know whether our response has addressed the concerns you raised. Your confirmation would mean a lot to us, as it will help ensure we’ve fully understood and addressed your valuable feedback.
> > > > >
> > > > > Thank you so much for your time and support throughout this process. We greatly appreciate your thoughtful input.
> > > > >
> > > > > Best,
> > > > >
> > > > > Authors

---

> ### Author Response · Authors · 2024-12-03
> **Urgent Reminder: Final Feedback Needed for Our Draft**
>
> Dear Reviewer Rg2d,
>
> As the submission deadline approaches, we wanted to gently remind you of the clarification we need regarding the concerns in your review. We've invested significant time and effort into polishing the draft, carefully addressing your suggestions.
>
> We understand you have a busy schedule, but your response is essential for us to proceed with the final version. Your input is invaluable, and we would greatly appreciate your insights at your earliest convenience.
>
> Thank you so much for your time and assistance.
>
> Best regards,
>
> Authors

---

### Official Review · Reviewer_srq7 · 2024-11-02

**Soundness:** 2
**Presentation:** 3
**Contribution:** 3
**Rating:** 6
**Confidence:** 3

**Summary:**

This paper addresses the challenge of identifiability in causal representation learning by examining how distribution shifts can be leveraged to uncover latent causal variables and their relationships. The authors establish a non-parametric condition that characterizes which types of distribution shifts contribute to identifiability within latent additive noise models, and prove partial identifiability results when only some distribution shifts meet this condition. They extend their theoretical findings to latent post-nonlinear causal models, making their framework more flexible. The authors translate their theoretical insights into a practical algorithm for learning reliable latent causal representations. Through experiments on synthetic data, image datasets, and fMRI data, they demonstrate their theoretical findings about identifiability.

**Strengths:**

- The paper provides complete and partial identifiability results - i.e. when not all distribution shifts adhere to Ass. (iv). This makes the results much more general and gives some hope for practical applications.
- I like the structure of how the theoretical results are introduced. First, the additive noise case is fully discussed. Then, the results are extended to the postnonlinear case. This reduces complexity for the main results and makes it easier to follow.

**Weaknesses:**

- This paper touches upon a CRL setting which is quite similar to the mutli-environment setting in [1]. One main difference that I can see is the assumed intervention type. However, the authors do not contrast their results to [1]. How is this work a generalization of that?
- The experimental evaluation is lacking many important details. E.g. for the synthetic data, the used mixing function $f$ is not stated. We know from prior work that recovering latents becomes harder with increasing complexity of $f$. Also, most or all (not familiar with all) baselines are misspecified regarding the problem due to assumptions of independent latents etc. In the experimental section, this is not explicitly mentioned, and it makes it seem like the proposed method is outperforming baselines that were developed for the same setting. Furthermore, no CRL method, such as e.g. [1] and others, is compared against. This makes it difficult to understand the practical applicability of the proposed approach.

**Minor:**

- L132: There seems to be a word missing.


[1] von Kügelgen et al. "Nonparametric identifiability of causal representations from unknown interventions." NeurIPS 2023

**Questions:**

- I'm confused about Assumption (iv): The setting is introduced as using soft interventions, however, as far as I understand, Ass. (iv) constrains the interventions to behave almost like a hard intervention. I.e. there exists an exogenous setting for each of the parents of a given variable such that the parents have no influence on the child. How is this assumption more general than a hard intervention?
- L160: Why do you only consider two parameter exponential family distributions? Is this a simplifying assumption, or is there some fundamental reason other distributions wouldn't work?
- L432: What is a method vs. a model?

---

> ### Author Response · Authors · 2024-11-23
> **Response**
>
> We thank Reviewer srq7 for taking the time to review our submission and for acknowledging that our results are easier to follow, more general, and offer hope for practical applications. Below, we address the reviewer’s concerns in detail.
>
> **Q1:** This paper touches upon a CRL setting which is quite similar to the mutli-environment setting in [1]. One main difference that I can see is the assumed intervention type. However, the authors do not contrast their results to [1]. How is this work a generalization of that?
>
> **R1:** Thank you for your careful review. We have incorporated reference [1] in the updated version. As noted in [1], it includes Assumption 2.9 (Perfect Interventions), which corresponds to hard interventions, the primary focus of most existing works. This stands in contrast to the soft interventions investigated in our study. In the introduction, we emphasized a research direction that leverages distribution shifts for identifiability analysis. A key distinction among these works lies in determining which types of distributions are suitable for specific function classes, such as those involving hard or soft interventions. Soft interventions, in particular, offer greater flexibility by accommodating a wider range of self-initiated behaviors compared to hard interventions. Advancing the development of soft interventions for identifiability is thus crucial for the causal representation learning community and represents a significant step toward a more practical and comprehensive theory.
>
> **Q2:** The experimental evaluation is lacking many important details. E.g. for the synthetic data, the used mixing function is not stated......Furthermore, no CRL method, such as e.g. [1] and others, is compared against.
>
> **R2:**We have updated the details of the mixing function in the synthetic data experiments. Regarding comparisons, a key challenge lies in the lack of alignment in assumptions across studies employing interventions. For instance, some studies assume hard interventions (e.g., [1]), while others, like ours, focus on soft interventions. Model choices further differ, with some assuming linear latent spaces and others adopting nonlinear models. Additionally, while some works rely on paired interventional data, our study addresses unpaired data. These inconsistencies make fair comparisons difficult, posing significant challenges for most studies, including ours, to provide comprehensive and meaningful analyses. In a comparable setting involving soft interventions, such as the polynomial models in [Liu et al., 2024], the primary distinction lies in the assumed function class for latent variables. In our experiments, we use this work as a baseline for comparison, as highlighted in the results.
>
> **Q3:** I'm confused about Assumption (iv): The setting is introduced as using soft interventions, however, as far as I understand, Ass. (iv) constrains the interventions to behave almost like a hard intervention. I.e. there exists an exogenous setting for each of the parents of a given variable such that the parents have no influence on the child. How is this assumption more general than a hard intervention?
>
> **R3:** Please note that soft interventions allow for the assignment of potentially functional transformations to a variable, which can, in general, encompass the specific changes induced by hard interventions, such as through the application of Dirac delta functions.  Assumption (iv) imposes constraints on the function class, requiring it to include a special point u' where the partial derivative is zero. Importantly, this function class is not limited to such special points u', it also includes other points that differ from u'. As highlighted in Remark 3.3, prior works based on hard intervention rely **solely** on the special point u' for identifiability. In contrast, our results demonstrate that any point within the constrained function class—whether the special point u' or any other differing points—can be utilized for identifiability. This makes our assumption strictly more general than hard interventions, extending the applicability of the identifiability analysis.
>
> **Q4:** Why do you only consider two parameter exponential family distributions? Is this a simplifying assumption, or is there some fundamental reason other distributions wouldn't work?
>
> **R4:** As we mentioned, the two-parameter exponential family eliminates the need for assumptions about the dimensionality of latent causal or noise variables. More details can be found in "Sorrenson, Peter, Carsten Rother, and Ullrich Köthe. "Disentanglement by nonlinear ica with general incompressible-flow networks (gin)." arXiv preprint arXiv:2001.04872 (2020)."

---

> > ### Author Response · Authors · 2024-11-23
> >
> > **Q5:** What is a method vs. a model?
> >
> > **R5:** Method involves different inference methods, such as VAE, iVAE, and Polynimals (Liu et al., 2024), while the model refers to different generative models, each with varying dimensions of latent variables. We have updated this for clarification. Thanks for your review.

---

> > > ### Comment · Reviewer_srq7 · 2024-11-26
> > >
> > > I thank the authors for their detailed rebuttal.
> > >
> > > After reading through the other reviews and rebuttals and after taking a look at the changes made to the manuscript, I adapt my rating as follows:
> > >
> > > I find the experimental section not very convincing. In particular, I don't buy that other CRL methods could not be compared against due to misspecification (the chosen baselines in the paper are all misspecified in some respect). Nonetheless, I believe that this paper makes a reasonable contribution through finding identifiability results for a setting with environments that differ by soft interventions.
> > >
> > > Other reviewers have brought up (i) unrealistic assumptions on the data generating process (ii) lack of novelty in the proof technique and (iii) lack of novelty w.r.t. Liu et al. 2024. I'm fine with (i): it's not like CRL has been the most practically applicable field so far. However, I can't judge (ii) and (iii) since I'm not very familiar with the applied proof technique and the work of Liu et al. 2024.
> > >
> > > Therefore, I will increase my score to 6, but decrease my confidence to 3.

---

> > > > ### Author Response · Authors · 2024-11-26
> > > >
> > > > Dear reviewer srq7
> > > >
> > > > We sincerely appreciate the time and effort the reviewer has devoted to thoroughly evaluating our work, especially for taking the time to reassess it in light of our revised version.
> > > >
> > > > Best,
> > > >
> > > > Authors

---

### Official Review · Reviewer_y9GW · 2024-11-06

**Soundness:** 3
**Presentation:** 3
**Contribution:** 2
**Rating:** 5
**Confidence:** 3

**Summary:**

This work studies non-parametric identifiability of latent variables in causal representation learning based on distribution shifts encoded in a conditional 2-parameter exponential family for noise variables, an additive noise model among latents, and a smooth invertible "observation function" mapping latents and additional independent noise to the observations. It provides conditions for full identifiability based on a novel assumption and provides partial (component-wise) identifiability based on the same assumption for certain components.

**Strengths:**

Overall, the paper is well structured, relatively easy to follow, provides a good overview of existing work on what assumptions can be leveraged for identifiability in CRL, and is attempts to provide some intuition for the problem setting.

**Weaknesses:**

What I consider to be the main weaknesses are (a) a suspected lack of novelty ('suspected' because I could not properly diagnose this due to a lack of mathematical rigor in the proofs) and (b) limited applicability due to the strong untestable results.
Finally, I do not see relevant methodological advances even though these are claimed as a contribution (l.078-079). As far as I understand from section 5, the developed method is a straight forward variational approximation of matching the required marginals with a canonical L_1 sparsity regularizer.

(a) As far as I understand, assumptions (i)-(iii) in Theorem 3.1 are well established and the way they are used in this work is also predominantly based on known results (Step I in the proof of Thm 3.1 using Sorrensen et al., 2020). Step II essentially "weaves in" the additional step from n to z, which is invertible by the additive noise assumption (another strong assumption, see below). Finally, the main novel contribution of the submission is in leveraging assumption (iv), step III, which in my view in and of itself is not a substantial addition to the existing literature, especially since assumption (iv) appears extremely restrictive (see (b)).
I also have to questions regarding the proofs:
* Throughout the proof of 3.1 in the appendix, there are multiple references to some "hatted" function 'having to belong to the same function class' to transfer properties of a non-hatted function to a hatted one. I do not see, where this assumption(?) comes from and what it means on a technical level? This appears to be a crucial element in various proofs, e.g., in lines 888-889.
* I am not yet 100% convinced by the sufficiency proof (Thm 3.5). In particular, the authors state that 944 that the required equality for matching marginals *can* be true even when $J_{\Phi_{2,1}} has a non-zero value, but I don't see why such a situation must always exist within the problem setting? I am thinking of situations where the partial derivative of a given component may be 0 except for a compact set where the noise distribution puts zero mass. (It was only assumed to be finite everywhere, but not positive everywhere...) Then one certainly should find counterexamples to the sufficiency statement?

(b) A key consideration when it comes to the impact of this work is whether the made (partially untestable!) assumptions can be defended in interesting real-world applications.
* In this regard, the authors mention in line 214, that 'these assumptions have been verified to be practicable in [...] application scenarios'. First, I'm not sure what it means for assumptions to be practicable, I suppose they're defendable or justified? However, when looking at the cited works, I do not see how these works justify the made assumptions in practice?
* In l.65, it is mentioned that 'additive noise models are particularly powerful in modern deep learning'. I believe this is disguising that additive noise models are another (possibly restrictive) assumption. Their assumptions happen to fit well to L2 minimization for which we have powerful deep learning techniques, but as I understand it, there is nothing inherently 'powerful' about additive noise models besides them being a heavily constrained class of models that happen to often allow for feasible theoretical analysis and are compatible with our practical learning algorithms.
* Leaving aside assumptions (i)-(iii), which may have been justified better elsewhere that I am not aware of, asusmption (iv) also appears very restrictive to me. I appreciate the attempt to justify it in protein-protein interactions (l.220+). When following up on the references, I believe (Forbes & Krueger, 2019) somehow snuck in there accidentially (not related to PPI or small molecules at all)? While I agree with PPI inhibition through small molecules is a hot and important area for drug discovery and that it appears to fit assumption (iv) on the surface level, it glances over many subtleties such as full inhibition (to the level of a partial derivative being zero) is rather unlikely and some sort of modulation being more the norm, such small molecules typically having other effects on the mechanisms involved beyond a mere soft intervention on a single protein concentration, PPI not even remotely being compatible with the DAG or additive noise assumption, etc. In short, I see the appeal of having mathematically identified a necessary and sufficient (see above) condition for identifiability, but do not see the relevance of these findings. At the same time, in my view the results are not sufficient for a purely theoretical contribution towards the mathematical aspects of identifiability in CRL.
* Lines 216-217 mention that for Theorem 3.1 one does not need to know the $\ell$ the dimensionality of the latent variable. However, for assumption (ii) to hold, one *at least* needs to know that there are $2\ell + 1$ different values of $u$. (When $u$ is an environment indicator, I could easily imagine there only to be a handful of different ones.) Hence, some form of implicit knowledge about $\ell$ is required to even have a chance that the assumptions hold?

**Questions:**

Besides the questions worked into the weaknesses above, there are a few minor details:

* there are multiple capitalized letters where they shouldn't be (l.63,884,...)
* l.116: we explores
* l.142: "models elucidate the underlying processes" (uncommon wording?)
* l.175: ...modulated by as outlined
* l.213: We here consider unitizes ...
* l.235: exists a team bz_1...
* l.237: groundturth
* l.239: both lambda' and lambda' must belong...
* l.250: "The whole function cl"
* some missing articles, e.g., l.387: we formulate prior model as; l.888: we have point u_i'...
* l.382: re-parametric trick
* l.393: "we simply impose L1 norm, other methods may also be flexible" (unusual wording?)
* l.888: must to belong the same

---

> ### Author Response · Authors · 2024-11-23
> **Response**
>
> We thank Reviewer y9GW for taking the time to review our submission and for recognizing well structured, relatively easy to follow. We address the reviewer’s conerns below.
>
> ----
>
> **Regarding to Proof**
>
> **Q1:** Some "hatted" function 'having to belong to the same function class' to transfer properties of a non-hatted function to a hatted one. I do not see, where this assumption(?) comes from and what it means on a technical level?  This appears to be a crucial element in various proofs, e.g., lines 888-889 in the orignial version, which corresponds to lines 943-944.
>
> **R1:** The underlying logic is that if we constrain the function class in generative models, the same function class can be enforced in inference models to maintain consistency with the generative model, without introducing any additional assumptions. Specifically, the statement in lines 888-889 in the orignial version, we have a point $\mathbf{u} _ {i'}$,  so that $\frac{\partial {h _ 3(n _ 1,n _ 2,n _ 3,\mathbf{u}=\mathbf{u} _ {i'})}}{\partial n _ {2}}=0$, is an alternative expression of assumption (iv) regarding function class constraints in generative models, supported by Lemma A.3. Consequently, the same function class constraints can be applied in inference models, such as ensuring $\frac{\partial {\hat h _ 3(n _ 1,...,n _ 3,\mathbf{u}=\mathbf{u} _ {i'})}}{\partial n _ {2}}=0$. This idea is analogous to assuming an exponential family distribution for noise in the generative model, which allows us to enforce the same distribution in the inference model, or assuming that the mapping from latent to observed space in the generative model is invertible, enabling the enforcement of an invertible mapping in the inference model as well.
>
> **Q2:** I am not yet 100 convinced by the sufficiency proof (Thm 3.5). In particular, the authors state that line 944 (e.g., 998 in the new version) that the required equality for matching marginals can be true even when  don't see why such a situation must always exist within the problem setting.
>
> **R2:** As stated in lines 944 in the original version 'This implies that $J _ {{\mathbf{\Phi}} _ {2,1}}$ can have a non-zero value.' The rationale here is that this step is intended to demonstrate **unidentifiability**. In this context, showing the **existence** of a scenario where $J _ {{\mathbf{\Phi}} _ {2,1}}$ can take on a non-zero value is **sufficient to establish unidentifiability**. It does not require $J _ {{\mathbf{\Phi}} _ {2,1}}$ to always have a non-zero value.
>
> -----
>
> **Regarding to assumptions**
>
> **Q3:** In this regard, the authors mention in line 214, that 'these assumptions have been verified to be practicable in [...] application scenarios'. First, I'm not sure what it means for assumptions to be practicable, I suppose they're defendable or justified? However, when looking at the cited works, I do not see how these works justify the made assumptions in practice?
>
> **R3:** Assumptions (i)-(iii), as discussed, were originally proposed in the context of nonlinear ICA and have since been widely utilized in a variety of real-world applications. Although fully justifying these assumptions in practical scenarios remains challenging, their adoption has enabled the development of theories and methods that consistently achieve state-of-the-art performance in the cited works. This success provides strong empirical evidence that these assumptions are not only theoretically meaningful but also practically viable, at least within the scope of the settings and applications explored to date.
>
> **Q4:** additive noise models are another (possibly restrictive) assumption.
>
> **R4:** Thank you for your insightful feedback. We acknowledge that additive noise models indeed represent a specific and constrained class of models. Our original aim was to emphasize that, in contrast to the polynomial models in [Liu 2024], which depend on fixed terms like $z^2_i$ or $z^3_i$ to represent nonlinear relationships, the nonlinear component in additive noise models is often more versatile. For example, it can be modeled using complex structures such as multilayer perceptrons (MLPs) or other transformations commonly employed in modern deep learning frameworks. We have revised this section in the updated version to better articulate this distinction. We appreciate your careful review and constructive comments.
>
> **Q5:** I believe (Forbes  Krueger, 2019) somehow snuck in there accidentially (not related to PPI or small molecules at all)?
>
> **Q5:** Thank you for pointing out the citation typo. The correct citation should indeed be: Scott, Duncan E., et al. "Small Molecules, Big Targets: Drug Discovery Faces the Protein–Protein Interaction Challenge." Nature Reviews Drug Discovery, vol. 15, no. 8, 2016, pp. 533–550.

---

> > ### Author Response · Authors · 2024-11-23
> >
> > **Q6:** it glances over many subtleties such as full inhibition (to the level of a partial derivative being zero) is rather unlikely
> >
> > **R6:** While full inhibition is rare in biological systems due to the complexity and adaptability of cellular mechanisms, there are instances where full inhibition can be achieved under specific, controlled conditions, particularly with small molecule inhibitors that exhibit high affinity and specificity for their targets. For example, gene editing technologies like CRISPR/Cas9 can effectively 'knock out' a protein or gene, leading to complete inhibition. Similarly, receptor antagonists can achieve full inhibition by completely blocking the activity of a receptor. Further, note that condition (iv) emphasizes the existence of full inhibition to limit the function class. In other words, condition (iv) can be satisfied as long as full inhibition is possible, regardless of the probability of its occurrence. Even if the availability of observed data at the specific point u' is not guaranteed, other points in the function class can still be used to identify latent causal variables, as mentioned in Remark 3.3. This is a key distinction from previous works that rely on hard interventions, which require the specific point u' to be present in the observed data.
> >
> > **Q7:** PPI not even remotely being compatible with the DAG or additive noise assumption, etc.
> >
> > **Q7:** We acknowledge that, in most real-world applications, justifying the assumptions of DAGs and additive noise models is inherently challenging. However, it is crucial to recognize that advancing toward a truly general function class and cycle graph is a long-standing and ambitious research goal—one that cannot be achieved in a single study. This work may represent a foundational step in this ongoing journey. By generalizing previous results and laying the groundwork for broader applicability, it provides a valuable stepping stone toward the ultimate goal of achieving identifiability under more general assumptions.
> >
> > **Q8:** Lines 216-217 mention that for Theorem 3.1 one does not need to know the dimensionality of the latent variable. However, for assumption (ii) to hold, one at least needs to know ...
> >
> > **R8:** The key point is that we do not need to use such prior knowledge exactly to complete the proof of identifiability. A rough estimation, such as knowing that the number of environments is greater than 2l+1, suffices.
> >
> > -----
> >
> > Thank you once again for your careful review. We have addressed the minor details you mentioned and updated them accordingly.

---

> > ### Comment · Reviewer_y9GW · 2024-11-25
> > **Thanks for the replies**
> >
> > Thanks for the detailed clarifications.
> >
> > Q1: I think I get the reasoning behind the hatted functions, it just still remains a bit abstract to me. Maybe writing this out more technically, by actually giving symbols to the assumed function classes and examples of what they could contain (i.e., how they are defined and constrained).
> >
> > Q2: I think this makes sense, thanks a lot.
> >
> > Q3,Q6-8: I agree with many things stated by the authors. At this point, on the scale from "overly restrictive" to "practically useful", I still tend to come down closer to the "overly restrictive" side. I do see the individual points, like the (good!) examples of full inhibition. But in these specific examples I am then worried about the context still not being the right one for this setting (e.g., Q7). In Q8, I also still think that knowing it has to be at least 2l+1 is a strong form of prior knowledge, even though it's more forgiving than "knowing it exactly".
> >
> > Considering the rebuttal, I will update my score to a 5.

---

> > > ### Author Response · Authors · 2024-11-25
> > >
> > > Dear reviewer y9GW
> > >
> > > Regarding the gaps between the assumptions in this work and real-world applications, such as Q7 and Q8, we acknowledge that this aspect has not been addressed as effectively as we would have hoped.
> > >
> > > We sincerely appreciate the time and effort the reviewer has dedicated to thoroughly reviewing our work, particularly for carefully checking our proofs step by step. Thank you for your thoughtful feedback and attention to detail.
> > >
> > > Best,
> > > Authors

---

### Author Response · Authors · 2024-11-23
**General Response**

We would like to thank the reviewers for their careful review and valuable feedback. We have addressed each of the concerns raised and made corresponding revisions to the paper. All updates and modifications have been detailed in the revised version, and we believe these improvements will enhance the clarity and quality of the paper. Once again, we appreciate the reviewers' support and contributions to our work.

---

### Meta-Review · Area_Chair_xueT · 2024-12-19

**Metareview:**

Overall, the authors' rebuttal has resolved a number of technical concerns from the reviewers. However, several reviewers still have concerns regarding the contribution in comparison to existing work this submission builds on. Furthermore, while some technical concerns were addressed conceptually, the reviewing panel feels that these details still need to be spelled out rigorously in mathematical language in the appendix to be verifiable. Ultimately, the rebuttal has not been able to fully resolve the reviewers' concern and lacking sufficient enthusiasm of this work, I unfortunately cannot recommend acceptance.

**Additional Comments On Reviewer Discussion:**

There has been little interaction between authors and reviewers. During the AC-reviewer phase, I have been able to follow up with the reviewers and get further clarification on their original assessment. Overall, most reviewers confirm that some of their concerns were resolved conceptually but these details still need to be concretely fleshed out in rigorous math. Without that, the reviewers do not feel confident to recommend an acceptance. Furthermore, there is a concern regarding comparison to existing works which this submission builds on. The central concept of "soft intervention" used in this paper is also not well articulated and might add confusion to the paper. Given this, I feel that this paper will indeed benefit from a thorough revision. Hence, my recommendation is a rejection.

---

### Decision · Program_Chairs · 2025-01-22

Reject